

# GeoDS (v.1.0) : a simple Geographical DownScaling model for long-term precipitation data over complex terrains

Jean-Baptiste Brenner[1], Aurélien Quiquet[1], Didier Roche[1], Didier Paillard[1], and Pradeebane Vaittinada Ayar[1]

[1]Laboratoire des Sciences du Climat et de l'Environnement, LSCE/IPSL, CEA-CNRS-UVSQ, Université Paris-Saclay Gif-sur-Yvette, France

**Correspondence:** Jean-Baptiste Brenner (jean-baptiste.brenner@lsce.ipsl.fr)

**Abstract.** Global climate models offer the most comprehensive description of the climate system and its internal processes to date but current computational capabilities typically restrict their spatial resolution to the order of tens of kilometers when multi-decennial or longer simulations are required. For climate applications, it is notoriously difficult to generate high spatial resolution data over long timescales (typically millennial). Over the years, various downscaling techniques have been developed

to generate fine scale data from climate models outputs but they often exhibit important limitations when applied over long periods of time. Building on previous efforts, we present a simple topography-based model (GeoDS) to downscale precipitation fields in complex areas, adapted to paleoclimate studies involving multi-millenia simulations. With a limited amount of inputs from a climate model and high resolved geographical information, the model computes, for each time step and every grid point, a topographic exposure index used to distribute precipitation into a high-resolution spatial grid. This dimensionless quantity

represents the exposure of surfaces to dominant windward incoming airflows, assumed to bring most of the humidity, and only depends on large scale winds and terrain configuration. The model is first tested under current climate conditions over part of the European Alpine region due to the availability of field data for comparison ; the complexity of both regional topography and climate conditions making it a good test of the proposed methodology. The relative effects of the model's parameters are assessed as well as the capacity of GeoDS to reproduce the spatial precipitation distribution of a well-resolved gridded target

dataset. Despite uncertainties regarding the correct wind fields to choose as input, and the dependency of the model to the temporal resolution of the large-scale data to downscale, we show that the procedure is able to capture most of the patterns occurring at fine spatial scale while being computationally inexpensive. We also demonstrate that the physical base underlying our work grants the model valuable robustness when used outside the calibration framework. This notably opens promising prospects for the application of GeoDS in paleoclimate contexts while providing a flexible, open source and well documented

downscaling tool for the climate community.





# 1 Introduction

Paleoclimate studies are valuable resources for improving our understanding of the mechanisms driving natural climate variability, as well as for reducing uncertainties associated with the effects of the current climate change. Although natural archives offer precious information to reconstruct past environmental conditions, limitations regarding their spatial and temporal coverage often lead to combining them with data derived from models (Ludwig et al., 2019). Among these, Global Climate Models (GCM) and Earth System Models, provide the most comprehensive representation of the climate system and the interactions between its components (Intergovernmental Panel On Climate Change (Ipcc), 2023). However, due to high computational costs and despite promising prospects offered by emerging supercomputing platforms, the horizontal resolution of such models currently remains restricted to 50-100 km (Schär et al., 2020) when multi-decadal or longer simulations are required, as it is typically the case for climate analyses. GCMs are thus too coarse to provide a realistic representation of key processes occurring at regional and local scales and yet essential to accurately depict paleoenvironments. In order to refine global models outputs, downscaling methods have been developed over the years and classically fall under one of the two main following categories. Firstly, dynamical downscaling (DDS) consists in explicitly resolving the regional atmospheric physics and dynamics. This can be achieved either by zooming over the studied area using a global climate model with flexible spatial grid (Hourdin et al., 2006) or by forcing a regional climate model (RCM) with output fields from a GCM (Jacob et al., 2014). Such limited-area models are supposedly able to spatially and temporally refine the global circulation signal by physically accounting for sub-grid processes and effects (e.g. mesoscale dynamical topographic induced phenomenons, distance to coastlines and inland water bodies) instead of the parametrisations used by the driving GCM. However, inner representations of physical processes proper to each model can lead to a potentially wide range of outcomes when applying a set of RCMs over the same region (Vautard et al., 2021). Besides, DDS generally involves elevated computational costs, limiting domain extent of RCMs as well as their usage over long periods of time, as commonly required for paleoclimate studies (i.e. multi-millennia simulations). Often used either as an alternative or a complement to DDS (e.g. bias correction steps or hybrid methods, (Boé et al., 2007), statistical downscaling (SDS) constitutes the second dominant disaggregation approach typically employed to generate high spatial resolution climate fields. Besides simple spatial interpolation procedures such as inverse distance weighting, spline interpolation or kriging used in geostatistics (Daly, 2006), SDS gathers various techniques like transfer functions, weather generators, weather typing among which analogs and model output statistics (Vaittinada Ayar et al., 2016), and more recently deep learning techniques (Sun et al., 2024). SDS does not physically represent the climate processes at play over the region of interest, but attempts instead to identify statistical relationships and correlations between coarse resolution variables or predictors, and local variables designated as predictands, usually the climate variables needed at finer scale (Vrac et al., 2012). Although highly cost-effective tools, these methods are extremely dependant on the spatial distribution and quality of observations used for calibration, hampering their use in regions with sparse station coverage such as mountainous areas (Fiddes and Gruber, 2014). Additionally, SDS relies on a strong hypothesis of stationarity, supposing that the statistical relationship built on the observation period (generally 50 to 150 years) remains valid through time (Schoof 2013 ; Ludwig et al. 2019). While this is usually not tested, this assumption is acceptable for recent past and present but may not stay valid under very





different climate conditions (Karger et al. 2023 ; Arthur et al. 2023). In areas with pronounced topography, an alternative category of spatial DS focusing specifically on climate-relief interactions can be used to obtain high resolution climate data fields. Orography constitutes a major driver of local climate, susceptible to produce highly changing conditions over short distances (Daly, 2006). For instance, average temperature gradients of several degrees can typically occur between north and south fac-
ing slopes in complex terrain, due to topographic exposure and the related differential heating of surfaces (Aalto et al. 2017 ; Böhner and Antonić 2009). Similarly, windward orographic uplift of incoming moist air masses and the associated rain-shadow effect on the leeward slopes often leads to strong variability of both temperatures and precipitation patterns in mountainous areas, with wetter and colder conditions on the windward side of the massif (Smith, 1979). Regionalisation procedures designated in the rest of this paper as geographical downscaling (GDS) typically encapsulate such effects by using terrain-based
information, such as elevation, slope or aspect, derived from high resolution Digital Elevation Models (DEMs). Faced with the diversity of existing methods, choosing one over the others must reflect the users needs and adapt to the constraints of the study. For long-term simulations under changing climate conditions, many topography-based models developed over the years present significant limitations. Statistical techniques, like the widely used Parameter elevation Regression on Independent Slopes Model (PRISM, Daly et al. 1994) which interpolates rain-gauge station data with elevation and aspect correction, are
mostly applied under recent past conditions due to their need for well distributed observational data. More physically based parametrisations of sub-grids orographic processes, like the Elevation Ratio Weighted Method or the Elevation Range with Maximum elevation Method (Tesfa et al., 2020) are flexible and computationally inexpensive, but rely solely on elevation to downscale precipitation. They thus neglect essential topographical signals like multi-scale leeward-windward gradients. Finally, other more complex approaches, such as the linear model of orographic precipitation developed by (Smith and Barstad,
2004) which includes basic elements like airflow dynamics or advection, notably improve the physical description of the effects of relief on climate. However, due to the temporal and spatial scale they solve, they often require an offline or online coupling with the model to downscale and are more costly in computer-hours than previously mentioned techniques. More recently, Karger et al. (2017) proposed a downscaling algorithm (CHELSA) based on empiric fine-scale terrain descriptors, combining local topographical information and large scale climate signals, to produce global temperatures, precipitation and derived
bioclimatic parameters climatologies at kilometric scale. Using terrain-based indicators that can be updated with an evolving orography (e.g. paleoclimate simulations) and independent of observations allows building robust relationships through time at low cost compared to dynamical methods, and constitute an approach of great interest for long term climate simulations at high resolution, even in a context of changing climate. A noteworthy application of the CHELSA algorithm under past climate conditions is the transient downscaling of temperatures and precipitation since the Last Glacial Maximum (Karger et al., 2023).
Despite promising results, the method neglects important aspects of orographic precipitation like the drying effect affecting interior regions of broad mountain ranges (e.g. the Alps, the Tibetan Plateau, Pepin et al. 2022), and lacks flexibility regarding parameters of the model as well as a detailed description of their relative effects. From a more practical perspective, the method is included in a Geographical Information System (SAGA-gis), which is not widely used within the paleoclimate modelling community (Willmes et al., 2016). This paper presents a simple alternative method to downscale precipitation fields in complex
terrain inspired from the philosophy underlying Karger et al.'s work and adapted to climate simulations over long periods of



time. Although the algorithm generates surface temperatures data as well, we focus solely here on total precipitation which exhibit the highest variability in mountainous areas and represent the greatest challenge to accurately predict (Tapiador et al., 2019). Choosing these variables as core targets for our downscaling model was motivated by several reasons, among which : 1) they constitute key components of local climate that can directly affect ecosystems and human activities : producing reliable temperatures and precipitation fields is of major interest, especially in a context of changing climate 2) their spatial distribution is highly conditioned by surfaces characteristics, in particular topography 3) an accurate description of their fine scale patterns is required by numerous models as input data, including those classically used for paleoenvironments reconstructions (e.g. ice sheet and vegetation models). In this study, we test and evaluate our model under recent past conditions due to the quality and coverage of observational data. Although daily data were available, we chose to work on a monthly basis. This is because most large-scale model outputs in paleoclimate contexts - which motivated to a large extent the development of this method - are available on a monthly time scale (e.g. Paleoclimate Modelling Intercomparison Project data). As for the studied area, we chose the Alps, since the massif exhibits a wide diversity of terrain configurations and climate conditions. Section 1 introduces the different datasets used in this study, while section 2 presents the method used to downscale precipitation fields. In section 3, we analyse the model outputs in various contexts, and we conclude with section 4, dedicated to the discussion and perspectives.

## 2 Data

### 2.1 Precipitation

#### 2.1.1 Target

We employed the Alpine Precipitation Grid Dataset (APGD) developed at MeteoSwiss (Isotta and Frei 2013 ; Isotta et al. 2014) and covering the entire alpine region (4.8-17.5°E / 43-49°N) over the period 1971-2019 as a proxy for observations. This openly available dataset uses measurements from more than 8500 rain-gauge stations combined with successive statistical interpolations and correction steps to provide robust estimates of daily precipitation sums (mm) over a regular 5km Cartesian grid using a Lambert Azimuthal Equal Area projection. The choice of this specific dataset compared to other similar products (e.g. EOBS which covers a longer time period) was motivated by its high stations density over the studied area, the thorough quality controls applied to the data coming from different contributions and the efforts made to reduce the risk of systematic underestimates at high elevations. Although given on a 5km scale, it is yet worth mentioning that the APGD effective resolution for daily totals is coarser, between 10 and 20 km depending on the density of stations. As previously mentioned, we worked for this study on a monthly basis. Our high resolution target APGDm was obtained by aggregating daily precipitation data to monthly sums.

#### 2.1.2 Coarse resolution climate data

Our model GeoDS (Geographical DownScaling) requires large scale precipitation and winds fields as inputs to generate local data. To simulate precipitation patterns over the Alps for the period 1971-2019 at coarse resolution, we degraded the APGDm



target dataset to a regular 50 km Cartesian grid using a first order conservative remapping from the Climate Data Operator package (version 2.4.4). This ensures spatio-temporal consistency between precipitation fields at low and high resolutions, and avoid conducting bias correction steps over GCM outputs. Since the APGD only provides precipitation data, we collected the eastward, u and northward, v wind components from the ERA-5 dataset (Hersbach et al., 2023), developed on a regular 31 km grid at the European Centre for Medium-Range Weather Forecast. In order to test the effect of wind height over the model's performances, several levels (10m, 950, 900, 850, 800, 700 and 500 hpa) were retrieved and degraded to the 50 km coarse grid. In order to avoid abrupt transitions between coarse grid points, we achieved a bilinear interpolation of large scale climate inputs (i.e. precipitation and wind fields) over the high-resolution topographic grid (sect. 2.2) as a preprocessing.

## 2.2 Topography

The local topographic information required to downscale precipitation fields were derived from the Shuttle Radar Topography Mission (SRTM) dataset produced by the National Aeronautics and Space Administration (NASA JPL, 2013). We aggregated elevation data from their native 30 sec grid to different target resolutions (Sect. 4). Elevations are expressed in meters above sea level.

## 3 Methods

To downscale long-term climate simulations at high resolution, our model has to meet certain requirements. Besides being computationally inexpensive, it needs to physically express the interactions between regional climate and surface topography, in order to limit its dependency on observational data and remain valid under changing conditions. The effects of relief on precipitation patterns are complex and, to propose a simple and robust method, we focused on modelling the first-order manifestation of orographic precipitation, marked by multi-scale windward-leeward gradients. When an air mass bearing moisture encounters a relief, it is forced to rise with the terrain. During the ascent, it undergoes an adiabatic expansion, decreasing its temperature as well as its capacity to contain humidity. If the condensation point is reached, clouds develop and water precipitates, usually on the windward slopes of the massif for large mountain ranges, nearer the crest for smaller hills. On the contrary, leeward locations tend to exhibit drier and warmer conditions, as airflows descent from the top of the relief and adiabatically compress. To take into account these effects, several working hypothesis underlie our work. In particular, precipitation patterns at high resolution are assumed to primarily depend on 1) the large scale signal, supplied by the model to downscale, which is supposed to be correct at its own resolution 2) the topographic exposure of surfaces to prevailing wind directions at each time step, considered to bring most of the moisture susceptible to feed precipitation. The detailed stages of the downscaling methodology are presented hereafter.





## 3.1 Topographic exposure of surfaces to incoming airflows

### 3.1.1 Topographic exposure index computation

The first step of our method consists in the derivation of a Topographic Exposure Index (TEI) from the elevation model and the large scale wind components. Each grid point in the studied area is assigned this dimensionless quantity at each time step. The indexes represent the surfaces' exposure to air masses coming from prevailing wind directions by taking into account previously crossed terrain's characteristics. A positive index suggests a relatively windward position compared to upstream points, while sheltered areas, like deep isolated valleys in complex environments, exhibit negative TEI values. Since exposure depends on airflow origin and pathway, different wind directions are expected to produce various index values for the same point. To cover all atmospheric configurations, TEI are thus computed for every main wind patterns addressed within the algorithm and defined as evenly spaced angular intervals between ]-$\pi, \pi$] (e.g. a [-$\pi/2, -\pi/4$] interval corresponds to a south-east airflow in an 8 directions configuration). For each of these prevailing wind directions, a list of windward positions is associated to every grid point M (x, y) of the domain by proceeding as follows :

1. The model first checks the distance between M and every other grid points in the domain. Only windward positions within a searching distance, noted dw_search, are retained for M's TEI calculations.

2. Each tested location is then sorted into one of the main airflow patterns depending on the principle angle formed by the horizontal x-axis and the segment between M and the tested location.

3. Finally, a relative weight is attributed to every tested grid point based on their horizontal distance to M. A near position, assumed to have more impact on M's exposure, will be given a greater weight compared to a location at the edge of the searching distance.

The number of prevailing wind directions, as well as the windward searching distance, are both model's parameters that can be modified within the configuration file. Thus the user can find the balance between performance and precision that meets his requirements. Note that we do not recommend setting too low values for neither of the parameters (i.e. not less than 6 wind directions and 30 kilometers searching distance based on our tests, sect. 4), since the induced computational gains come with heavy degradations of the model's performances. Figure 1 shows how windward positions are selected using different parametrisations.

Once grid points of influence have been identified for every main airflow patterns, the topographic exposure index of M is computed using (1) :

$$TEI_w(M) = \frac{1}{n} \sum_{k=1}^{n} \frac{h(M) - h(P_k)}{d(MP_k)} \tag{1}$$

where $TEI_w(M)$ is the topographic index of $M$ for the prevailing wind direction $w$, $n$ the number of windward positions $P_k$ within the searching distance influencing $M$'s exposure to the corresponding $w$ airflow pattern, $h(M)$ and $h(P_k)$ the





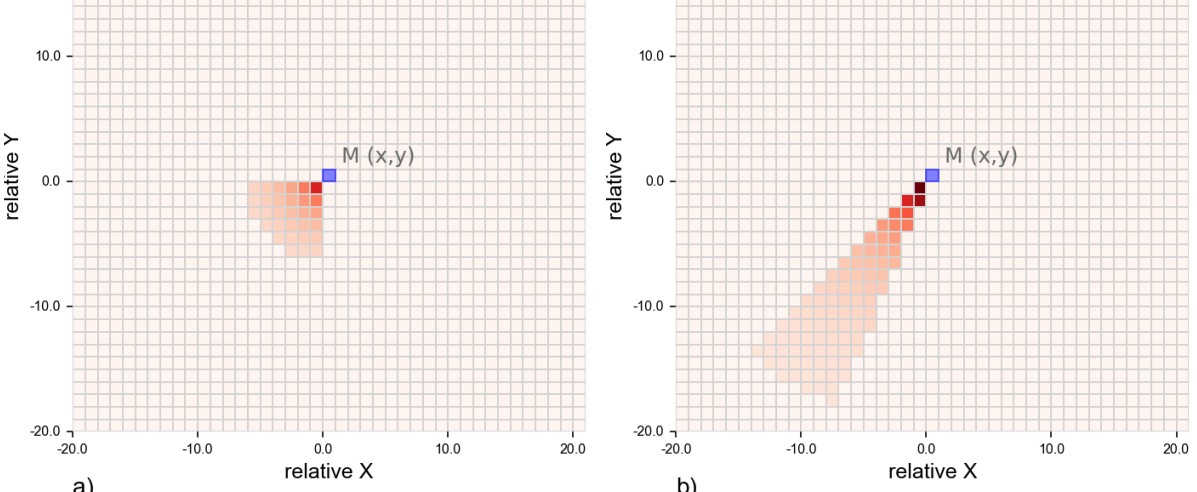

**Figure 1.** Selection and weight attribution of windward positions affecting the topographic exposure of any point M(x, y) for a) a four prevailing wind directions and 35 km windward searching distance configuration b) a sixteen prevailing wind directions and 100 km windward searching distance configuration. Increments in X and Y are given for any spatial resolution (a 5 km DEM in the above case)

respective elevations of $M$ and $P_k$, and $d(MP_k)$ the horizontal distance between $M$ and $P_k$, with $h(M)$, $h(P_k)$ and $d(MP_k)$ expressed in the same unit. Since the number of windward positions influencing $M$'s TEI increases with the input DEM's resolution, dividing by n ensures consistency when computing exposure indexes from elevation models at various scale. Thus, the mismatch between TEI's values at different resolutions is only caused by the added topographic information the finer scale DEM provides, all other variables being equal.

### 3.1.2 Drying effect correction

At this stage of the procedure, TEI overall exhibits a positive altitudinal gradient. Due to our working hypothesis, this tendency directly reflects on the precipitation patterns at high resolution, with precipitation rates also increasing with elevation. Although this is generally true (Daly, 2006), several exceptions exist, including tropical reliefs for which precipitation rapidly decreases above condensation level around 1000-1500m, and wide mountain ranges like the Alps or the Tibetan Plateau (Böhner et

Antonic, 2009 ; Gerlitz et al. 2015). In this case, most of the impinging precipitation occur at the edges of the mountain range, leaving inner areas relatively dry, despite higher elevations (Pepin et al. 2022). To take into account this drying effect, a correcting term can be computed for every grid point of the domain by checking the exposure indexes of positions upstream from the tested cell. Since positive TEI is associated with higher precipitation, the model simulates a strengthened drying effect as the number of exposed positions windward of the tested surface increases. This assumption illustrates the idea that

windward locations, if well exposed to air masses, tend to reduce water availability for downstream positions by draining the limited water resources stored within the atmosphere. The correction is first computed using equation (2) :



$$TEI\_drying\_effect\_correction_w(M) = \frac{1}{m} \sum_{l=1}^{m} TEI_w(P_l), TEI_w(P_l) > 0 \tag{2}$$

where $TEI\_drying\_effect\_correction_w(M)$ is the TEI drying effect correction of $M$ for the prevailing wind direction $w$, $TEI_w(P_l)$ the topographic indexes of the $m$ windward positions $P_l$ influencing $M$'s exposure within the $dw\_search_{drying\_effect}$.

The correcting term is then applied using (3) :

$$TEI_{w,corr}(M) = TEI_w(M) - \gamma * TEI\_drying\_effect\_correction_w(M) * \frac{h(M)}{h_0{}^3} \tag{3}$$

where $TEI_{w,corr}(M)$ is the drying effect corrected TEI of $M$, $h(M)$ the elevation of $M$ and $h_0$ an elevation of reference ($h_0$=1m) used to ensure TEI remains dimensionless and $\gamma$ a dimensionless parameter that must be calibrated. We apply an altitudinal factor cubed to enhance the drying effect over the highest locations where the model tends to be excessively humid

according to our tests. The user can choose to activate or not the above correction depending on the studied area. To account for the fact that the drying effect can affect precipitation patterns in complex environments over considerable distances, positive TEI are retrieved within an equal or longer searching radius than the one used in (1). A comparison of TEI maps over the Alps computed with or without drying effect is shown in Figure 2.

The topographic exposure indexes strongly depend on the windward searching distance used for their calculation. Short

$dw\_search$ values (Fig. 2.a) causes the TEI to exhibit marked gradients in complex regions, as it captures local terrain configuration information. Conversely, increasing this distance smooths TEI distribution (Fig. 2.b). Averaged over a larger area, the indexes offer in this case a more integrated description of the topography and its effects on surfaces exposure to airflows. Activating the drying effect correction allows to reduce TEI values over elevated locations (Fig. 2.c). Our tests showed this correction can help better representing precipitation patterns over inner regions of the massif (Appendix A Figure A3). Finally,

The comparison of Figure 2.a and 2.d illustrates that topographic indexes exhibit consistent patterns and range of values when computed using input DEM at different resolutions. For efficiency purposes, the entire procedure used to compute TEI intervenes within the algorithm as an initialisation step. Instead of calling the different routines at each time step, all indexes are calculated once for every grid point and each prevailing wind direction. The exposure information is then integrated to the rest of the downscaling method as detailed hereafter.

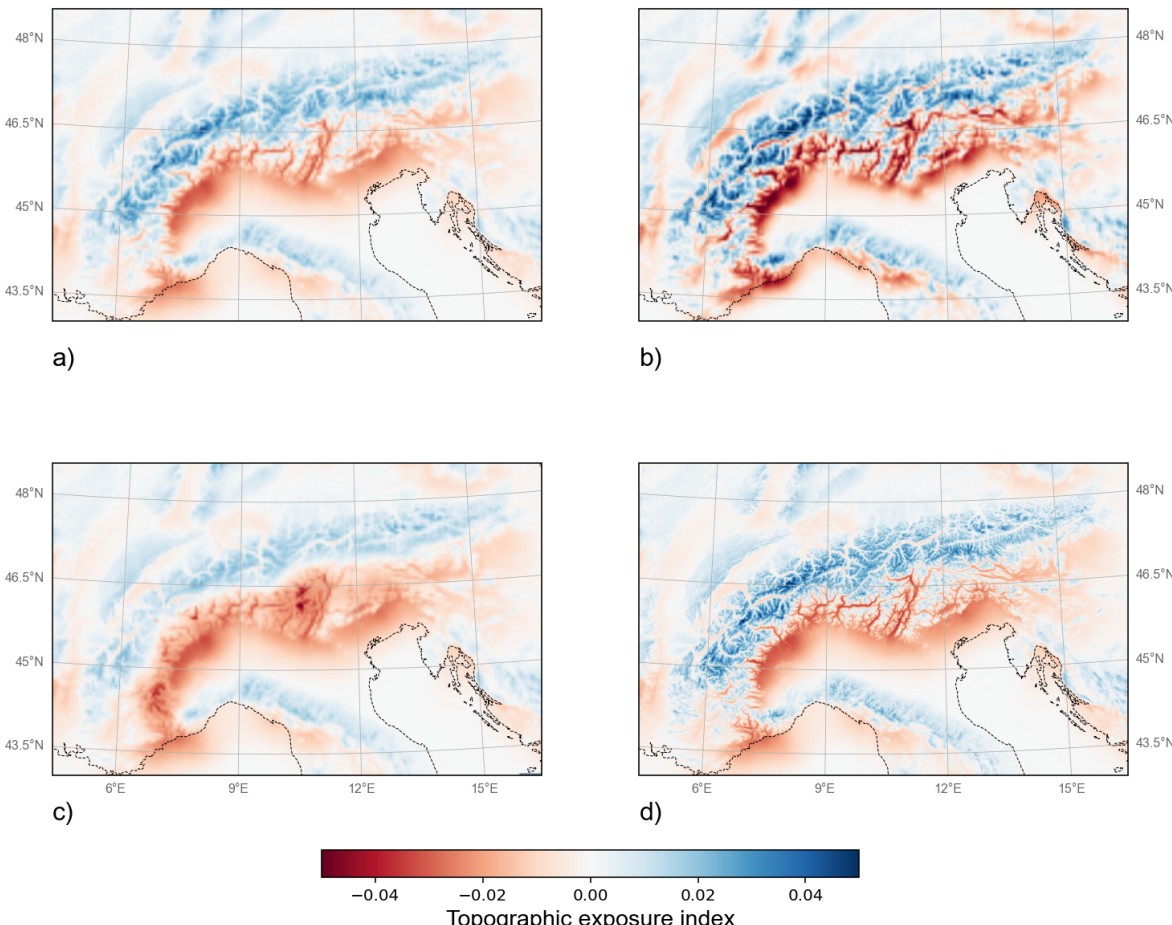

**Figure 2.** Topographic exposure indexes maps over the Alps computed for a north-west prevailing wind directions and different parametrisations : a) a 100 km windward searching distance, with no drying effect activated, b) a 50 km windward searching distance and no drying effect activated, c) a 100 km windward searching distance and the drying effect activated with a 150 km searching distance d) a 100 km windward searching distance and no drying effect but using a 1km DEM instead of a 5km elevation model (a, b, c).





## 3.2 Precipitation downscaling

Downscaled precipitation fields result from the combination of GCM precipitation, prevailing wind patterns over the domain and topographic exposure indexes of the surfaces. Once TEI for all possible configurations are computed, the algorithm associates at each time step and to every grid point $M(x,y)$ one of the major airflow directions $w$ (Sect. 3.1.1), based on the angle formed by the vector $(\boldsymbol{u} + \boldsymbol{v})$ and the horizontal x-axis, with $\boldsymbol{u}$ and $\boldsymbol{v}$ the wind components in the x and y directions at M. Then, high resolution precipitation are calculated as follows :

$$P_{HR}(x,y,t) = P_{CR}(x,y,t) * min(e^{\beta * TEI_{w,t}(M)}, max\_P\_increase\_factor) \tag{4}$$

where $P_{HR}(x,y,t)$ is the downscaled high resolution precipitation received by $M$ at time step $t$, $P_{CR}(x,y,t)$ the coarse resolution precipitation interpolated to the fine grid and received by $M$ at time step $t$, $\beta$ a dimensionless coefficient to calibrate, $TEI_{w,t}(M)$ the topographic exposure index of $M$ for the wind direction $w$ prevailing at $t$ over the surface of coordinates $(x,y)$, and $max\_P\_increase\_factor$ the maximum ratio of $P_{HR}/P_{CR}$. Equation (3) converts the topographic information added by the high resolution DEM into a modulation of the large scale atmospheric signal. At each time step, windward locations, i.e. with positive TEI, experience increased precipitation compared to the low resolution, while sheltered areas can only receive a fraction of what the GCM simulates. The prevailing wind directions update at each time step and every grid point allows for a dynamical repartition of precipitation through time, since exposed surfaces for a given airflow pattern at time $t$ can turn into absolute leeward positions at time $t+1$ after a shift in the wind's pathway. The exponential formulation ensures that $P_{HR}$ remains positive, even for the most sheltered locations of the domain, while mimicking the Clausius-Clapeyron relationship between the saturation pressure vapor of water and the temperature (As TEI increases, typically with elevation, precipitation increase as well ; similarly, as temperature decreases with elevation, the capacity of the atmosphere to contain water decreases, until saturation occurs). As for the upper limit of $P_{HR}$, an *ad hoc* ($max\_P\_increase\_factor$) parametrisation avoids any mathematical excessive enhancement of $CR$ precipitation. The value of this top $P_{HR}/P_{CR}$ ratio, corresponding to the maximum by which topography can increase low resolution precipitation, can be changed by the user.

## 4 Results

### 4.1 Sensitivity to the parameters of the model

#### 4.1.1 Statistical analysis

First, we tested the sensitivity of our model to the different parameters $\beta$ (dimensionless coefficient linking TEI and precipitation), $\gamma$ (dimensionless coefficient associated to the TEI drying effect correction), $nbr\_wdir$ (number of prevailing wind directions), $dw\_search$ (windward searching distance used to compute TEI) and $max\_P\_increase\_factor$ (maximum quotient between downscaled and coarse resolution precipitation). We used the Latin hypercube sampling method (randomLHS function from the lhs package version 0.16) with R (version 4.0.3) to generate an ensemble of 1500 parametrisations and





downscale on a monthly basis (588 time steps) the 50km resolution precipitation fields (Sect. 2) to the native APGD 5km

grid with GeoDS. Then, we compared each simulation to the target dataset and computed two quality metrics : 1) the Mean

Absolute Error (MAE) corresponding to the absolute differences between predicted and target precipitation data, averaged over

all grid points and time steps ; 2) the Mean Absolute Error over quantiles ($MAE_{quantiles}$), i.e. the average magnitude of the

absolute errors between the simulated and target precipitation quantiles, build over all grid points and time steps. Since the

$MAE$ corresponds to the mean error of the model at the scale of the grid point, it is sensitive to the spatial distribution of the

modelled data, while the $MAE$ over quantiles informs on the ability of the model to simulate the global statistical distribution

of the target. Results are presented in Figure 3.

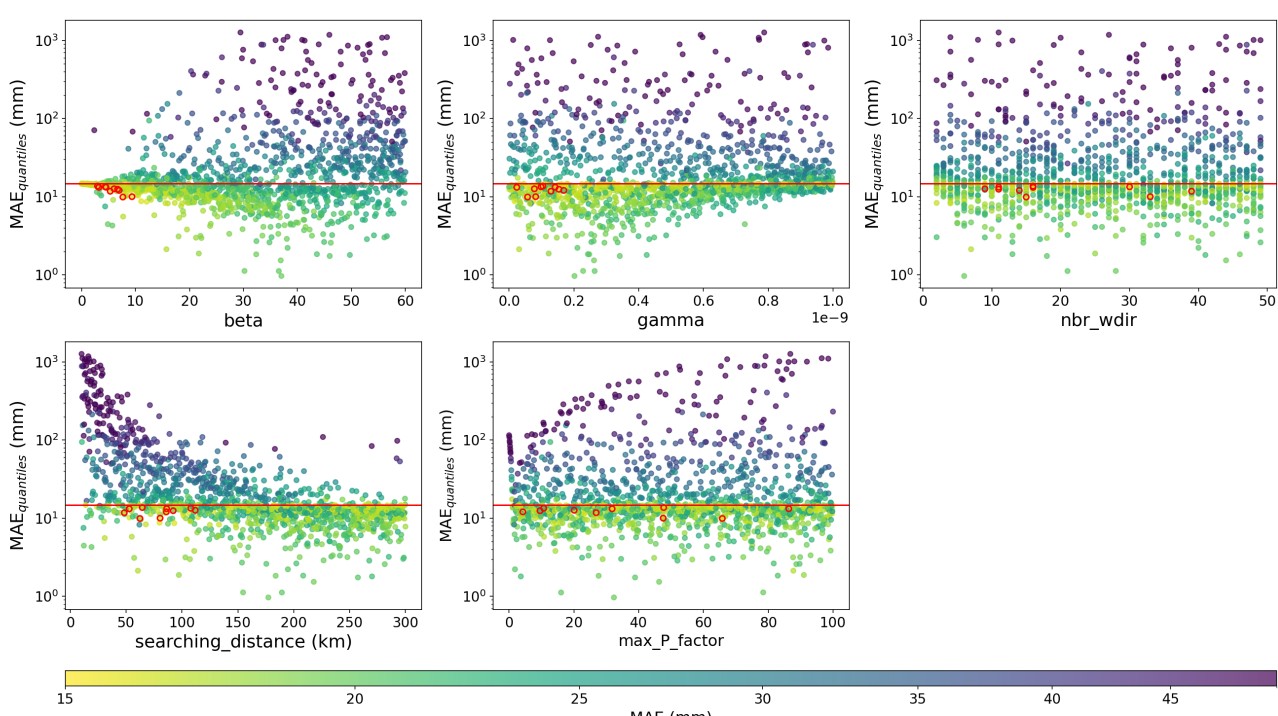

**Figure 3.** Quality metrics for the ensemble of 1500 simulations generated with GeoDS using different parametrisations. The y-axis indicates the $MAE_{quantiles}$ while the colour shading represents the $MAE$. Both quality metrics are expressed using a logarithmic scale. The red lines indicate to the $MAE_{quantiles}$ of the $REF_{interpolated}$ dataset. The red circles highlight the 10 simulations with the lowest $MAE$.

     The model appears to be primarily sensitive to $\beta$, $\gamma$, and $dw\_search$, both quality metrics varying with the tested values

of each parameter. Conversely, $nbr\_wdir$ and $max\_P\_increase\_factor$ seem to have more limited effect on GeoDS per-

formances, since a parametrisation limiting the model biases can be found for every value of these two variables. If choosing

small searching distance in the windward direction degrades both quality metrics, the repartition of the colours (and the red

circles) suggests that using low values for the two first parameters is best to reduce the $MAE$. This corresponds to a situation





where the effect of topography is limited or even suppressed : in this case, the model simply applies a bilinearly interpola-
tion to large-scale data. However, low values of $\beta$ and $\gamma$ don't optimise the $MAE_{quantiles}$, which notably informs on how
well extreme values of precipitation are represented. Accurately representing the highest precipitation levels is, however, of
paramount importance, as they strongly influence global water balance over the domain. The behaviour of the model when
$max\_P\_increase\_factor$ varies suggests that the model is able to regulate itself. This is particularly important for the maxi-
mum precipitation increase factor, which was implemented as an ad hoc limit to ensure precipitation downscaled with GeoDS
always remain in a plausible range (Sect. 3.2). As for the number of prevailing wind directions, although it appears to be of
limited impact, it illustrates that the sole statistical analysis of the $MAE$ and $MAE_{quantiles}$ is insufficient to conclude over
the role of the different variables, as we show hereafter.

### 4.1.2    Effects on precipitation patterns

The previous sensitivity test is useful to better apprehend the relative importance of the different parameters. For each one,
it also gives a valuable insight of the range of value to chose from to calibrate the model but it remains incomplete to select
the parametrisation that best fits the target. For instance, a set of parameters may produce a good statistical distribution of the
data, but fail to capture precipitation spatial patterns. To evaluate the capacity of GeoDS to generate accurate high-resolution
climate fields, we analysed, for different parametrisations of the model, the spatial distribution of downscaled precipitation
compared to the target. For these specific simulations, we also quantified the added value provided by the topographic model
compared to a reference noted $REF_{interpolated}$. This dataset simply corresponds to the 50km input climate fields (Sect. 2.1.2),
bilinearly interpolated to the target resolution. We deliberately chose to evaluate our model against an interpolated product
rather than the raw 50km input dataset (Appendix A Figure A1). Since the first step of our algorithm involves the same bilinear
regridding, the comparison to $REF_{interpolated}$ ensures that any potential improvements along the downscaling process is only
due to the topographic information the model adds. From the analysis of precipitation annual climatologies calculated over
the whole period, we were able to formulate several observations. Note that we could not study in detail the precipitation
patterns associated with every possible configurations of GeoDS, but visually analysing the outputs of our method, even for a
limited number of simulations, helped to understand the role and importance of each parameter of the model. Spatial patterns
are strongly degraded (smoothing effect) when setting $\beta$ too low (< 10) while high values (> 25) lead to overestimating the
impact of topography on precipitation, with exaggerated gradients between lowlands and summits. Small values of $\gamma$ don't
significantly affect precipitation distribution but the drying effect it applies over elevated locations notably increased as we
used higher coefficients. Consistently with the previous statistical results, short searching distance (< 30km) also degrade
spatial patterns, while increasing this variable led to smoother local gradients. For the $nbr\_wdir$ variable, our tests showed
that the number of prevailing directions had to be set in average greater than 6 to obtain consistent spatial distribution of
precipitation with the target, while higher values had primarily an impact on the statistics. As for $max\_P\_increase\_factor$,
it has little influence on precipitation patterns, corroborating the fact this artificial boundary is not necessary for the model to
perform well, if the parameters with the strongest impact are adequately chosen.





## 4.2 Comparison of downscaled precipitation to the native APGD

We analyse hereafter the downscaled dataset that offers the best spatial agreement with the target's precipitation patterns, while maximising the different quality metrics, i.e. MAE and $MAE_{quantiles}$. We also computed the dimensionless coefficient of determination $R^2{}_{quantiles}$ of the linear regression (slope=1, intercept=0) linking the target precipitation quantiles to the

predicted quantiles. The closer $R^2{}_{quantiles}$ is to 1, the better the model reproduces the target quantiles. These results were obtained using the ERA-5 700 hpa wind fields and the set of model's parameters shown in Appendix A Table A1. Locations mentionned hereafter are placed on the 5km DEM of the area available in Appendix A Figure A2 to facilitate the results analysis.

Figure 4 shows the distribution of the mean annual precipitation (1971-2019) over the Alps for the three versions of the
APGDm (coarse fields reinterpolated, downscaled and target datasets ; Appendix A Figure A3 presents the same data but visualised as anomalies of precipitation between the target and the predicted fields). As mentioned above, the objective hereafter is not to analyse in detail the precipitation patterns occurring in the Alps, but to focus on the performances of the downscaling model. For an exhaustive description of the native APGD climatology, please refer to Isotta et al., 2014. Although the effective resolution they resolve is different, the $REF_{interpolated}$ and HR datasets capture some key characteristics of the climate at the
scale of the whole massif. In particular, they both reproduce the elongated wet anomaly over the northern rim of the mountain range. Enhanced precipitation are also distributed on the southern slopes of the Alps, with a first wet zone south of Lucerne (Switzerland) along the St-Gottard section and covering the Italian Alps north of Milan, and a second area extending from the Dolomites to the Julian Alps in northern Slovenia. Besides flatlands, both datasets also exhibit a major dry anomaly in the inner Alps (region of Südtirol, Italy), also visible on the target. As pointed out in Frei and Shär (1998), these patterns are
widely driven by the main topographic features of the domain, and result from the typical interactions between large scale circulation and orography (Sect. 3) that can be captured even at relatively low resolution. An exception worth mentioning concerns medium mountain areas, like the Jura and the Vosges, as well as the western limits of the Alps. In these regions, the $REF_{interpolated}$ dataset significantly underestimates windward sloping precipitation, while our model is able to notably reduce the dry biases inherited from the input low resolution climate. Most of the added information GeoDS provides yet intervenes
at smaller scales, of the order of a few tens of kilometers and below. The downscaling model not only better represents the strong gradient between pre-Alps lowlands and outer slopes of the massif (e.g. region of Lausanne, Switzerland, region of Lombardia, Italy), it also reveals essential topographic features the $REF_{interpolated}$ dataset fails to capture. This concerns especially deep highly dry valleys, like the region of Grenoble in France, the valley of Sion in Switzerland or the region of Feldkirch in Austria. By taking into account the terrain exposure to dominant winds and due to the steepness of surrounding
slopes, the model well reproduces the rain-shadow effect largely responsible for the observed low precipitation levels in these areas (e.g. around 750 mm/year in the valley of Sion, against 1186 mm/year in average over the whole domain). GeoDS is also able to better simulate local orographic uplift and increasing precipitation with altitude in several locations, like the region of the Pizzo di Coca in Lombardia or the region of the Triglav summit in Slovenia. Overall, the downscaling procedure leads to a notably better description of topography-induced precipitation patterns and partially corrects the smoothing effect of local



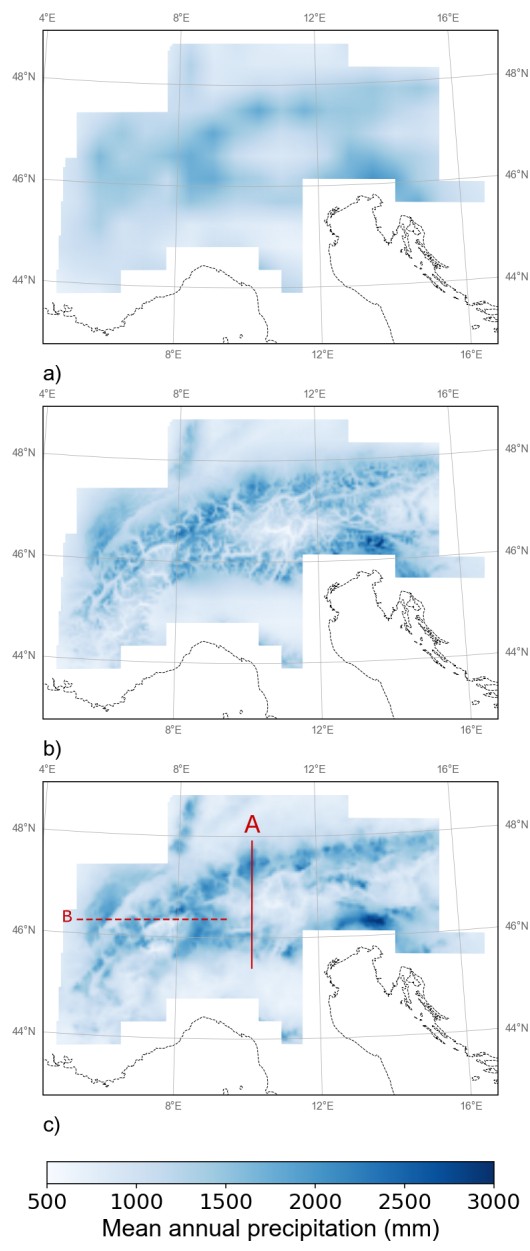

**Figure 4.** Average annual precipitation for the bilinearly interpolated $REF_{interpolated}$ (a), the downscaled DS (b) and the native high resolution (c) versions of the Alpine Precipitation Grid Dataset over the period 1971-2019 using a 5km input DEM. A indicates the location of the latitudinal cross section presented in Figure 5, while B indicates the location of the longitudinal cross section shown in Appendix A Figure A4



gradients exhibited by the $REF_{interpolated}$ simulation. Although simple and neglecting complex small scales phenomenons (Sect. 5.4), it provides a good first order description of the climate-orography interactions not only at the scale of the whole massif but also locally.

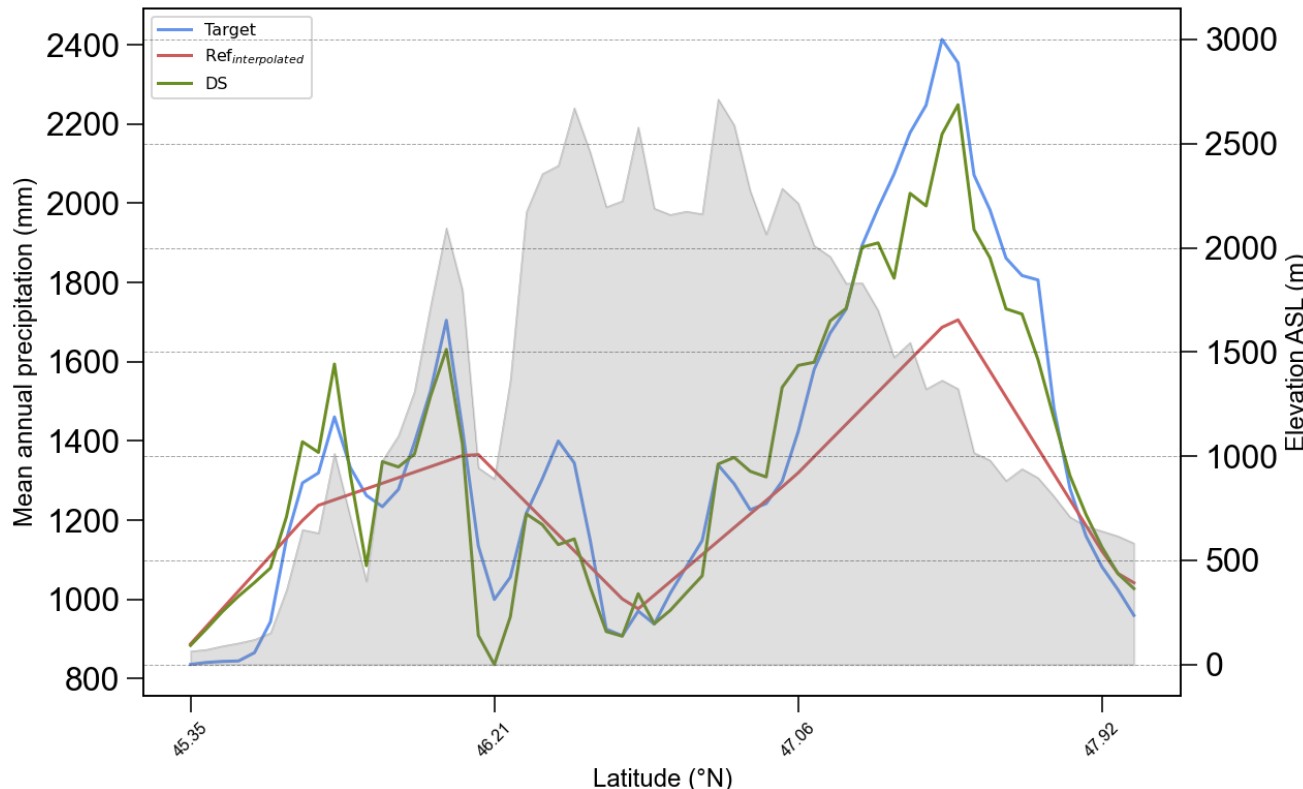

**Figure 5.** Mean annual precipitation (colored lines) and topographic elevation (filled grey curve) along a latitudinal section across the central Alps (A segment, Figure 4.c).

A complementary visualisation of the improvements brought by our model is shown on Figure 5. This north-south cross-section well illustrates the windward/leeward effect previously discussed. Oceanic westerly and north-westerly moist air masses
are forced to rise with surface elevation, regularly until reaching their condensation level. This leads to higher precipitation on the windward slopes while leeward sides remain relatively dry. At the scale of the whole massif, the rain-shadow effect on south facing slopes is partially compensated by Mediterranean air flows bringing moisture to the Italian Alps. Note that inner areas of the mountain range exhibit significantly lower precipitation levels despite higher elevations, due to the drying effect discussed in section 3. Our model generates notably more accurate precipitation patterns along the transect compared
to the $REF_{interpolated}$ simulation. An additional West-East cross-section is available in the Appendix A Figure A4. Previous conclusions regarding GeoDS performances are based on the visual analysis of precipitation spatial distributions. The statistical



examination of the quantile-quantile plots and the associated quality metrics of the $REF_{interpolated}$ and DS datasets presented in Figure 6 corroborates herein-above results.

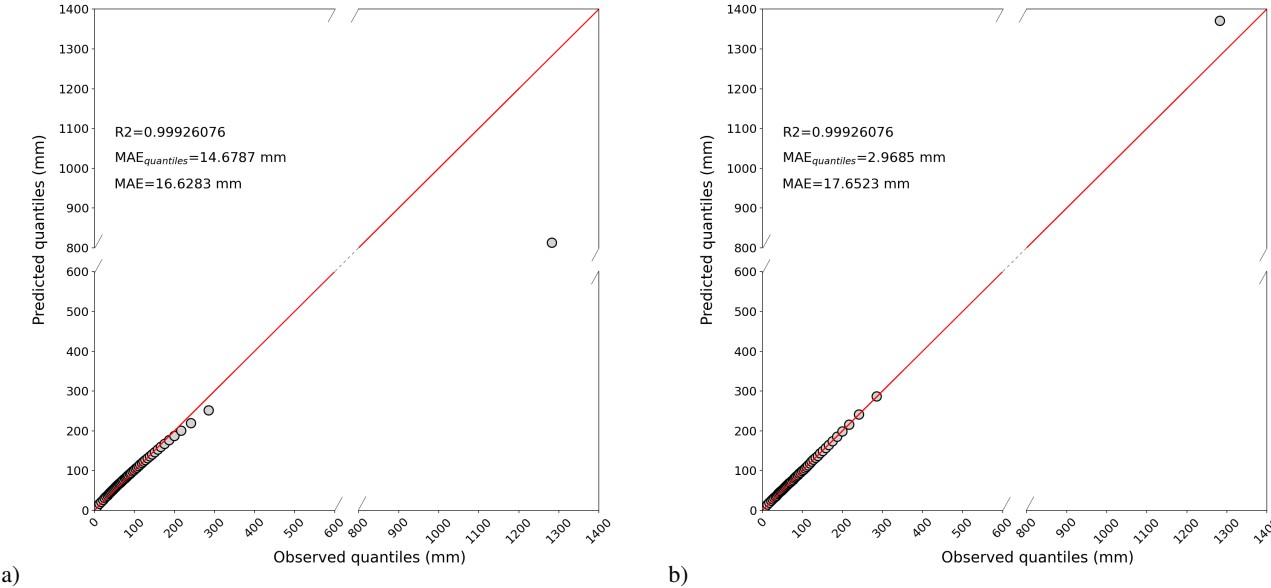

**Figure 6.** Comparison of observed monthly precipitation quantiles and a) Refinterpolatedb) DS monthly precipitation quantiles using a 5km input DEM. For clarity concerns, the distribution was divided into 50 regularly spaced quantiles. Note that we used broken linear scale to represent all quantiles.

The $REF_{interpolated}$ dataset shows a good agreement with the target, but slightly overestimates precipitation levels under
90 mm/month, and notably underestimates the values of the highest quantiles (>150 mm/month). By comparison, the DS simulation fits significantly better the native APGD dataset. The $R^2_{quantiles}$ increases from 0.978 to 0.999, while the mean absolute error over quantiles is reduced from 14.7 to 2.9 mm/month. Overall, the dry bias affecting the highest precipitation levels is well corrected by our algorithm. This is consistent with previous observations : the detailed topography of the area allows the model to enhance precipitation over elevated surfaces compared to the $REF_{interpolated}$ dataset, reflecting the
orographic uplift of incoming air masses above the condensation level. Similarly, the downscaling procedure accounts for the related rain-shadow affecting leeward slopes and deep valleys, and regularly reaches lower precipitation levels than the reference. This reduces the overestimation of low precipitation quantiles and contributes to improving overall results. Although globally performing well, several limits to our approach must be pointed out. Besides the major dry bias over the Julian Alps and the region of the Lago Maggiore, Italy (discussed in sect. 5), significant discrepancies between downscaled and target
climates arise at very fine scale ($\approx$ 5km), where the model tends to exaggerate local windward/leeward effects compared to the target. In such areas, simulated strong local gradients bring out small defined topographic features, like micro-scale dry



valleys and ridges, while the native dataset exhibits more homogenous patterns over elevated complex terrain. Examples can be found in the northern parts of the Italian region of Piemonte and Lombardia, as well as in the northern Austrian Alps. The overestimation of the impacts of topography on precipitation distribution is assumed to be unrelated to the input DEM high

resolution (5km), since orographic effects, including windward-leeward slopes gradients, can occur at finer scales (Daly et al., 1994 ; Smith, 1979). Instead, the mismatch between downscaled precipitation fields and the target are more likely due to 1) the model's shortcomings, discussed later on and/or 2) the effective resolution of the APGD. Although given on a 5km grid, the actual scale resolved by the dataset is in the order of the inter-station spacing, between 10 to 20km depending on the area (Isotta et al. 2014). Therefore, small scale topographic features, like isolated valleys at high altitudes with low station coverage,

may be not well represented by the dataset. Since the APGD is considered the best proxy for observations in our study, we tried to test our second hypothesis and to better match the native dataset by using a coarser 10km resolution DEM. The results are shown in Figure 7.

Running GeoDS with a 10km input DEM leads to a better spatial agreement between simulated and target precipitation distributions. As expected, the degraded topography smooths very fine scale patterns. This helps to correct wet anomalies over

different areas (e.g. the Nesthorn region) although it exacerbates specific dry biases, notably over the Slovenian Alps. Statistics confirm the visual analysis with a mean absolute error over quantiles reduced from 2.97 to 1.73 mm when using a coarser resolution for the input DEM. For the rest of the results presented in this paper, note that we used a 10km elevation model to describe high-resolution topography.





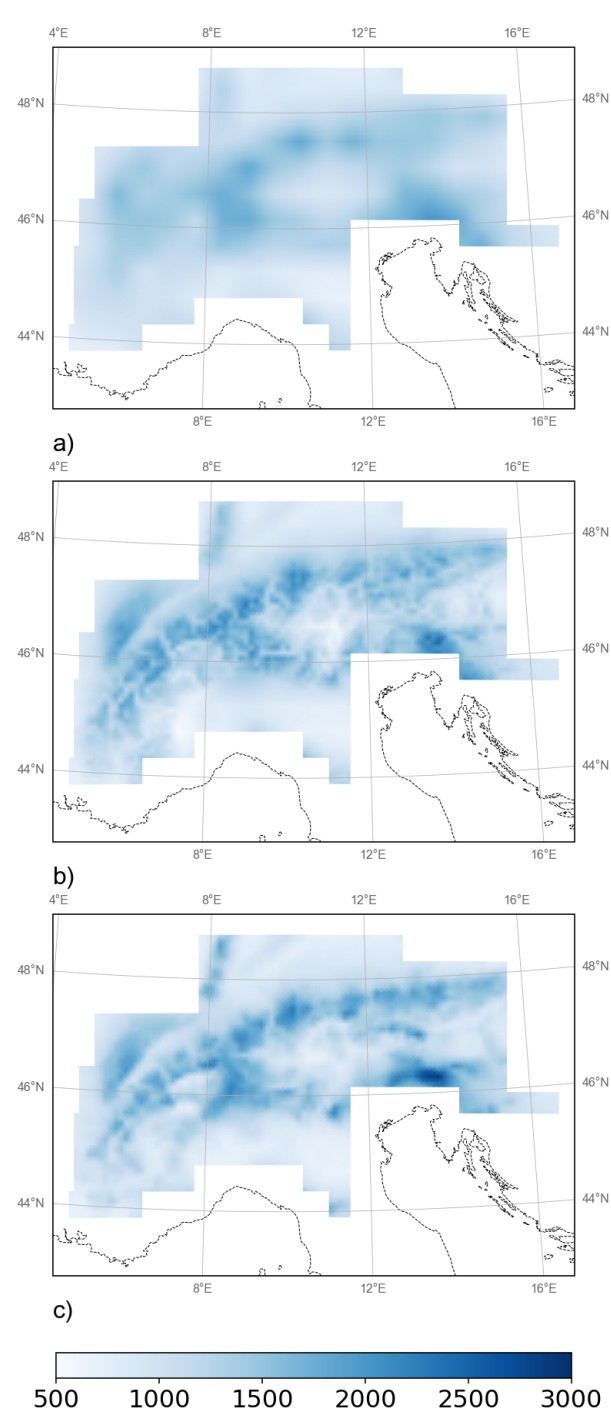




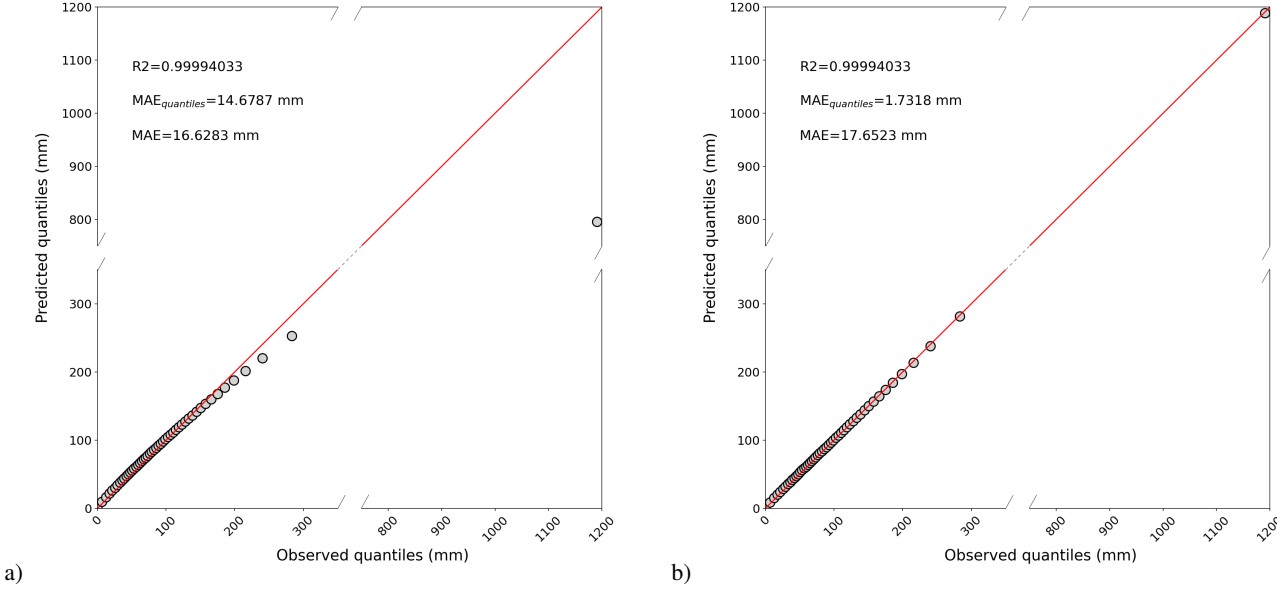

**Figure 7.** 1) Average annual precipitation for the coarse resolution (a), the downscaled (b) and the native high resolution (c) versions of the Alpine Precipitation Grid Dataset over the period 1971-2019 using a 10km input DEM. 2) Comparison of observed monthly precipitation quantiles and a) Refinterpolated b) DS monthly precipitation quantiles using a 10km input DEM.



To conclude this section, it is important to keep in mind that previous analyses only concern temporally averaged data. Although GeoDS produces satisfying climatologies and improves global statistics, compensatory phenomenons can occur over time and mask poor temporal agreement with observations. This is yet an essential criteria to take into account for various applications (e.g. monthly accumulation levels for glacier modelling). To verify how the algorithm behaves temporally, we calculated the climatological seasonal cycle over the whole domain and for several particular grid points located in various topographical contexts (see Appendix A Figure A5 for interannual variability). Results are shown in Figure 8.

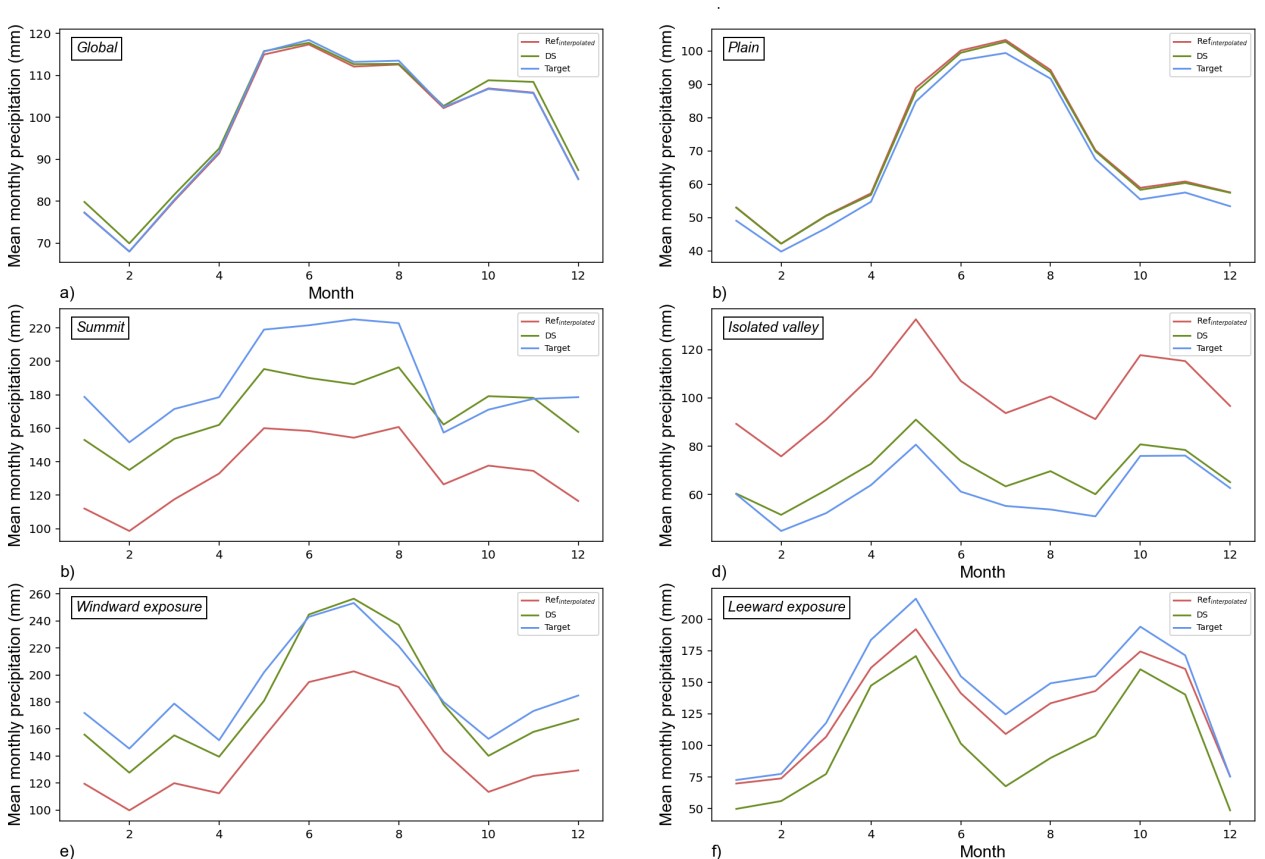

**Figure 8.** Mean target and predicted seasonal cycles of precipitation calculated over the period 1971-2019 on a global scale and for grid points located in different topographical contexts. The location of each grid point is shown in Appendix A Figure A2

On a global scale, the algorithm is slightly less accurate than the bilinearly interpolated simulation which, by construction, exhibits global mean values at each time step close to the target's (Sect. 2). The grid point labelled "Plain" we chose is located in a flat opened area. As expected, the difference between the DS and the $REF_{interpolated}$ datasets are thus very limited, since no significant topographical information is added along the downscaling process. For the windward and summit locations, the model better simulates how precipitation tend to increase with surfaces elevation, if not affected by the strong drying effect typically occurring over alpine inner lands. GeoDS also significantly improves the seasonal cycle occurring in deep valleys,





usually in the rain-shadow cast by surrounding relief and therefore notably dry. The grid point exhibiting on average a leeward exposure to incoming airflows is where the model performs worst. We selected a position in the region of the Lago Maggiore, protected from westerly average winds by the Alps and previously noted as particularly biased. Please refer to section 5.2 for a detailed analysis of this specific issue. Note that other grid points with leeward exposure exhibit much better agreement with the target (e.g. region of Geneva, Switzerland). Overall, these temporal analysis are consistent with previous observations and contribute to identify topographical contexts where the model performs best and where it generates the highest biases.

## 5 Discussion

In addition to the previous results, this section discusses several key aspects of our downscaling method and highlights some of its limitations.

### 5.1 Wind height effect

Our approach strongly depends on accurately identifying where most of the humidity feeding orographic precipitation comes from. In this study, we approximated the origin of moisture with the average wind direction, derived from the u an v wind components at a given level and computed at each time step and for every grid point in the domain. Not only this assumption is a strong working hypothesis worth discussing (Sect 5.3), but also, since the vertical structure of the atmosphere is not homogeneous, the direction of the mean wind can significantly vary from one level to the other. For instance, even if for most of the period westerly slopes of the massif are exposed to near-surface winds coming from the West and South-West, air fluxes at higher pressure levels often originate from the North-West, with potentially strong effects on precipitation distribution depending on the input fields used. At this point, it can thus be challenging to propose a systematic method for selecting the wind height maximising the model's performances. If, at the scale of the whole area, using the 700 hpa wind components offers the best global spatial and statistical agreement with the target, an average wind computed at 850 hpa improves for instance results over Northern Italy. Although a simple approach relying on a single pressure level, as the one used in section 4, is sufficient for the algorithm to perform well, future works should consider improving the description of the origin of moisture-bearing airflows.

### 5.2 Sensitivity to the spatio-temporal resolution of input data

So far, we used our model to downscale climate data that were aggregated from the target dataset on a 50km grid to limit large scale biases. Although this resolution is too coarse to take into account some essential features of local precipitation (Sect. 4), it gives a good overview of patterns occurring at the scale of the massif. In practice, many GCM are not able to supply such well-resolved fields. To evaluate how the algorithm performs when fed with coarser inputs, we built highly degraded large-scale climate data by averaging at each time step the APGD precipitation and the ERA-5 700 hpa u and v wind components over the whole domain. We then downscaled the resulting spatially homogeneous fields with our model. Unlike the 50km dataset which includes by construction some topographical information, these new inputs don't capture any effect of the relief on





precipitation patterns over the domain. Calibration factors thus need to be updated to avoid widely underestimated orographic precipitation by keeping parametrisation unchanged. Results discussed hereafter were obtained with the set of parameters (see Appendix A Table A1) offering the best spatial agreement with the target and optimising the statistical metrics presented in

section 4.

The downscaling procedure allows to reduce the $MAE$ over quantiles from 40.96 to 15.13 mm, while the R2 increases from 0.62 to 0.99. Despite overestimating the actual range of precipitation levels, the model is able to capture some key characteristics of their regional distribution. Enhanced precipitation are well represented over the Vosges, the Jura and the region of Genoa while strong transitions between plains and sloping surfaces clearly highlights the limits of the alpine massif. In

agreement with the target dataset, the drying effect incorporated into the model causes inner regions to exhibit drier conditions than outer slopes, despite high elevations. At a finer scale, the model reveals essential topographical features, like deep dry valleys (e.g. region of Grenoble, Aoste, Sion). Naturally, topography alone is insufficient to accurately represent the complexity of patterns occurring in the Alps. Large scale atmospheric circulation, distance to littoral, latitude or land cover for instance determine to a great extent precipitation spatial distribution, but their effects don't manifest at the sub-domain scale due to the

resolution of the input fields. This leads to major biases in several areas, like the Southern Alps which should be significantly drier because of the influence of the Mediterranean climate, or the northern slopes of the massif, actually much more humid than what the model simulates due to the exposure to moist oceanic air masses coming from the Atlantic Ocean and the North Sea. Note that the present test corresponds to an extreme situation, where one grid cell of a GCM covers the entire domain. Downscaling a model that includes a very basic North-South gradient and a coarse effect of the continent over only several

grid points, would likely lead to much better results.

If our model obviously depends on the spatial resolution of large-scale inputs, it may also be affected by their temporal resolution. As mentioned in section 3, GeoDS relies on the strong assumption that the average wind, calculated at each time step and every grid point, brings most of the impinging precipitation. Although it is true on a monthly basis for a large extent of the studied area, precipitation in several regions and for certain months are primarily due to short or isolated events. This

concern for instance the Southern Alps, subject to eastern return phenomenons, the Julian-Carnic Alps or the region of the Lago Maggiore, often affected by strong precipitation coming from the Mediterranean and the Adriatic seas (Frei and Schär, 1998). In these cases, the actual moisture-bearing airflows differ from the average wind used as input by GeoDS. However, the duration of such events leaves the associated change in the mean monthly wind directions widely undetected, and can lead to significant differences between the downscaled dataset and the target. To better capture patterns in areas affected by severe

rainfalls, we focused on northern Italy where GeoDS exhibits the strongest bias and we selected a month where the model performs particularly poorly. We then applied our downscaling procedure with daily inputs over the region and compared the resulting daily precipitation aggregated over the month to the initial monthly simulation and the APGD native target. Results are shown in Figure 10.

The dataset derived from monthly input data is significantly drier than the target. The westerly and south-westerly average

winds over the region cause the algorithm to diminish input precipitation due to the rain-shadow cast by the relief in the wind-ward direction. On the other hand, using daily input data allows to better represent the high precipitation levels characteristic




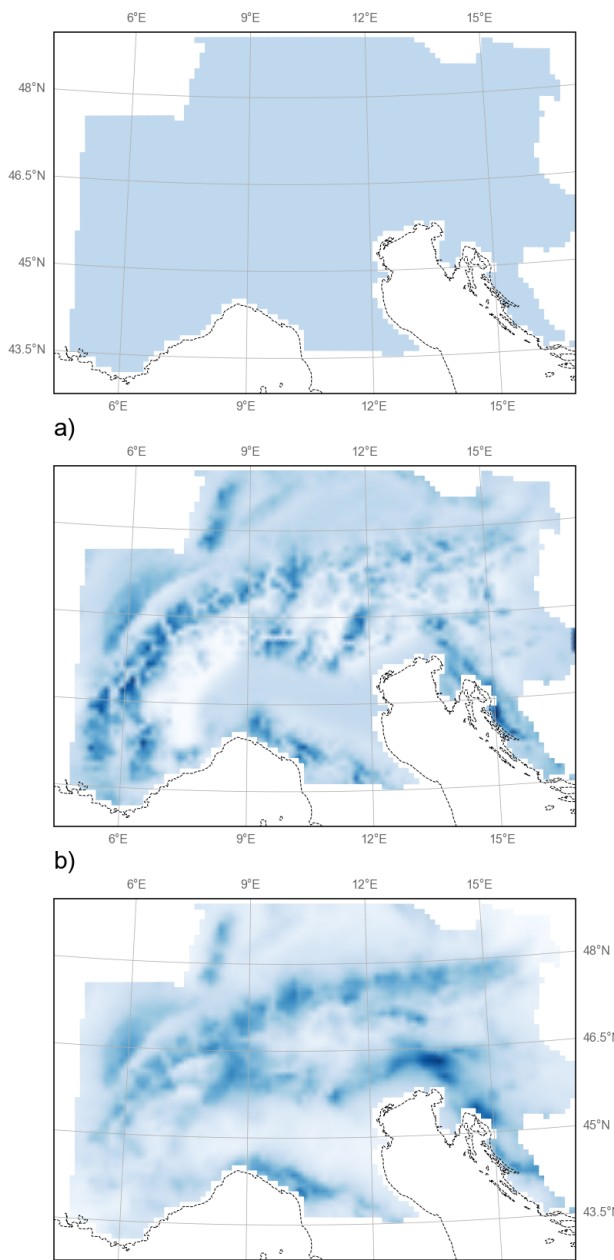

**Figure 9.** Average annual precipitation for the highly degraded resolution (a), the downscaled (b) and the native high resolution (c) versions of the Alpine Precipitation Grid Dataset over the period 1971-2019 using a 10km input DEM

of the Lago Maggiore and its surroundings. While airflows are still primarily coming from the west and south-west, the finer temporal scale ensures the model takes into account isolated events associated with eastern and south-eastern winds that bring most of the monthly moisture. In these cases, the relief acts as an enhancing factor rather than a barrier, thus reinforcing im-



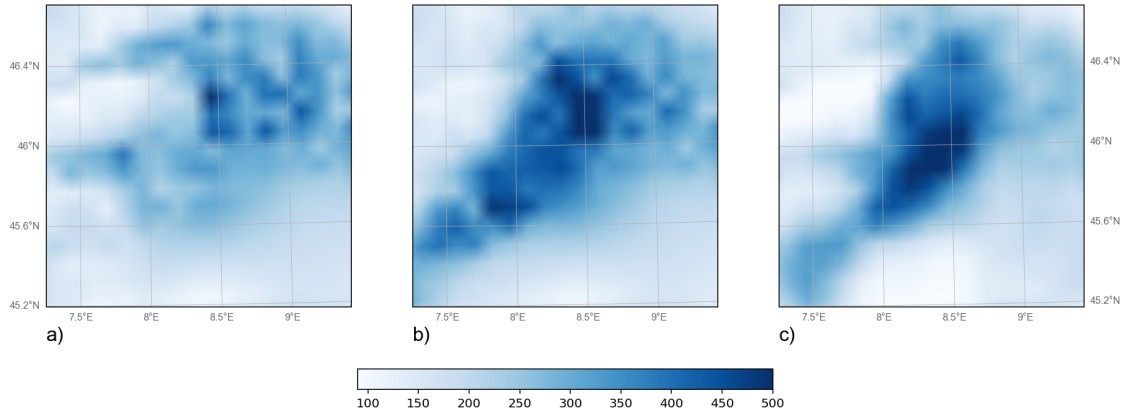

**Figure 10.** Precipitation over the region of the Lago Maggiore, Italy, in October 2019 for a) the GeoDS downscaled dataset using monthly input data, b) the GeoDS downscaled dataset using daily input data, c) the native APGD target.

pinging precipitation over the region. Although the gains highly depend on the region, increasing the temporal resolution of the model is a leverage to consider in order to improve GeoDS performances.

## 5.3  Robustness of the model

As we intend to apply GeoDS in various contexts, including for the downscaling of paleoclimate simulations, it is crucial to evaluate the robustness of our approach. To do this, we need to analyse, for different climate conditions and geographical

configurations 1) how well can the underlying theory of the model explain spatial distribution of precipitation 2) to what extent the optimal parametrisation of the algorithm for a given resolution and inherited from the calibration framework needs to be updated. This second criteria can be of major interest for applications involving limited observational data available to refit the model. To answer these questions, we used GeoDS to downscale precipitation fields over Greenland for the period 1990-2020. Climate of Greenland is very different from the Alps, with most of the region exhibiting very dry conditions,

except for outer slopes which receive most of the impinging precipitation, primarily coming from the South-East. The terrain configuration is also very distinct from the alpine orography, due to the ice sheet covering most of the domain and limiting strong inner elevational gradients and the presence of major local topographical features like deep valleys, except at the edges. Since observations over Greenland are very sparse and no equivalent product to the APGD exists, we used the precipitation fields generated by the MAR (Modèle Atmosphérique Régional) climate model and tuned over the domain as our fine scale

target. MAR is a regional physically based model (Fettweis et al., 2013), forced at the boundaries by reanalyses and often used to estimate ice sheet mass balance variations over Greenland (Shepherd et al., 2020). Note that data at our disposal were available on a 15 km regular grid, thus coarser than the alpine target dataset. After retrieving MAR simulations, we applied the same procedure as for the Alps. First, we degraded MAR to a 50km Cartesian grid using a conservative remapping to simulate the outputs of a GCM while limiting large-scale biases. While keeping the alpine parametrisation of the model

unchanged, we then downscaled precipitation using the ERA-5 $u$ and $v$ wind components at 700 hpa and remapped to the




50km grid, and the native MAR elevation data at 15km as inputs. Finally, we compared GeoDS outputs $DS_{Greenland}$ to the target and to a reference simulation noted $REF_{interpolated\_Greenland}$, obtained by a bilinear interpolation of the large-scale precipitation from 50 to 15km. Since precipitation levels are lower and local gradients not as marked over Greenland as in the Alps, we calculated the climatological anomalies between each predicted dataset and the target and applied a logarithmic scale

of colours to highlight extreme biases and facilitate the interpretation of results. Results are shown in Figure 11.

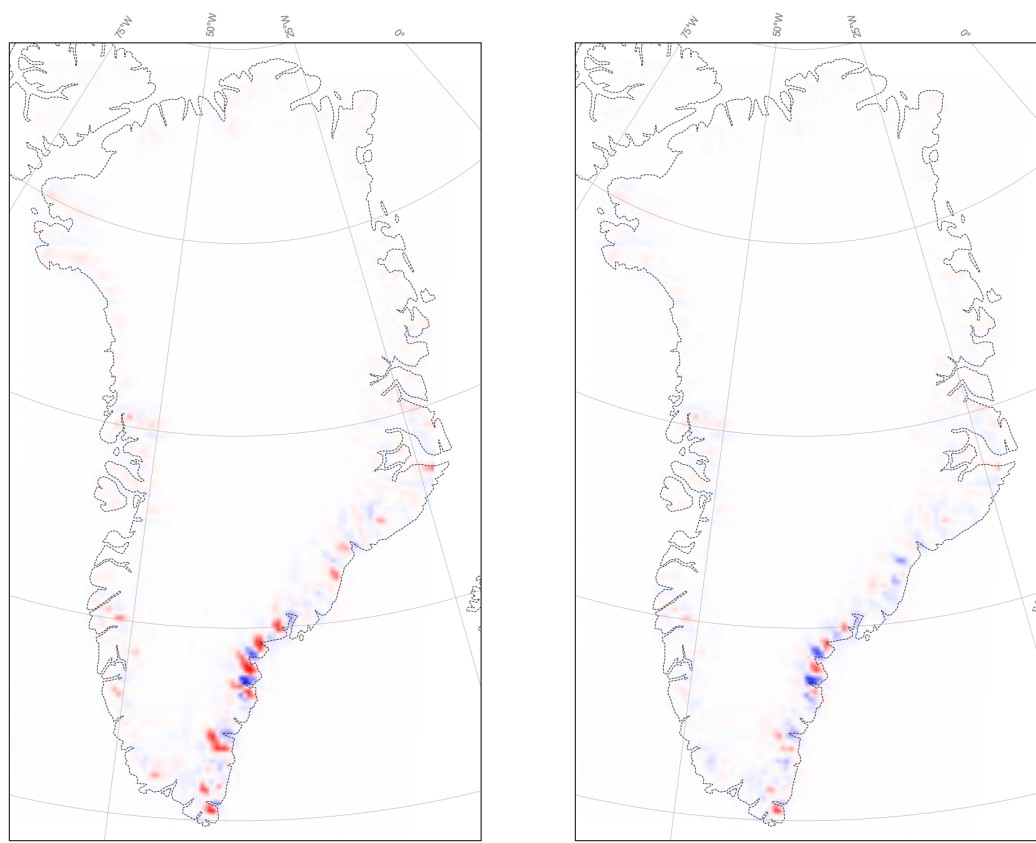

**Figure 11.** Mean annual precipitation anomaly between MAR and a) the $REF_{interpolated\_Greenland}$ b) the $DS_{Greenland}$ datasets, calculated over the period 1990-2020. Sub-figure c) shows the native MAR topography used as input DEM. Red (respectively blue) areas correspond to underestimated (resp. overestimated) predicted precipitation compared to the target

Both reference and downscaled datasets exhibit the strongest errors over outer slopes along the coast, where most of the precipitation occur. In these areas, the dominant effect of relief on climate modelled by GeoDS is the enhancement of impinging precipitation with elevation. This positive altitudinal gradient allows the algorithm to significantly reduce dry biases associated with the interpolated dataset. GeoDS yet fails to correct regions with excessive humidity : since the procedure can solely

diminish input precipitation in areas sheltered from incoming airflows by upstream barrier, or through the drying correction affecting deep inner regions, the resolution of the input DEM, the location of the target regions as well as the exposure of




the coastline to dominant winds limit compensatory drying effects. Despite these limitations, our method improves global precipitation distribution, and allows to reduce the mean absolute error over quantiles from 8.5 to 2.59 mm/year while the $R^2_{quantiles}$ increases from 0,995 to 0,999. It should be noted that we were able to obtain better results by varying the calibration coefficients around the optimal parametrisation inherited from the alpine tests (by switching $\beta$ from 16 to 14 and $\gamma$ from $5e^{-10}$ to $7e^{-10}$, we reduced the $MAE_{quantiles}$ to 0.69 mm/year). We yet choose to present here results generated with the model tuned over the Alps rather than Greenland to illustrate the robustness of the method.

## 6 General remarks and conclusions

The forced orographic-induced uplift of air masses leading to increased precipitation on windward slopes is a good first-order description of the interactions between climate and topography and explains most of the mesoscale precipitation patterns occurring in mid-latitude mountain ranges (Roe, 2005). However, this simple model is often insufficient to capture the complexity of orographic precipitation locally. Small scales patterns result from a combination of multiple factors, including stability and strength of the incoming airflow, particle growth microphysics, terrain height and geometry and thermodynamics of the impinging moist air (Houze Jr., 2012). These effects can significantly modify the locations where actual precipitation occur. For instance, stable atmospheric conditions and low wind speed can lead either to windward stagnation or leeward convergence of incoming air masses depending on relief elevation and width. Similarly, night-time cooling and the associated gravity flow can indifferently generate precipitation at the windward or leeward base of the mountain, if moisture conditions of low-level currents allow it (Houze et al. 2012). Solving fine scale physics and dynamics of the atmosphere is inconsistent with the underlying philosophy of our model. However, the limited large-scale inputs used to downscale precipitation within GeoDS contribute to the simplicity of our approach, but also likely limit its performances. Further works should consider integrating supplementary GCM variables and fine scale terrain-based descriptors, like the large-scale wind speed or the distance to major water bodies, as prospects for improving precipitation patterns at high resolution. Despite the various limitations of the model presented in this section, GeoDS meets several of our key initial expectations. Besides its flexibility, it requires limited inputs as well as low computational resources (the simulation using the parametrisation presented in Appendix A Table A1 takes less than 20 secondes using an Intel Core i7-1365U with 16Gb of RAM), while being able to capture essential precipitation patterns over a region exhibiting both complex topographical environments and various climate conditions. Finally, the method showed relative good robustness when applied outside the calibration area, opening up promising prospects for paleoclimate applications.

*Code and data availability.* The GeoDS algorithm is written in Fortran 90 and freely available on Zenodo under the Apache-2.0 license (https://doi.org/10.5281/zenodo.17045252, Brenner 2025a). The archive contains the source code as well as input data (adapted from Isotta and Frei (2013) and Hersbach et al. 2023) to test the model and regenerate outputs presented in the article. The downscaled datasets with GeoDS presented in the paper are stored as well under a Zenodo archive (https://doi.org/10.5281/zenodo.16420097, Brenner 2025b).





**Appendix A**

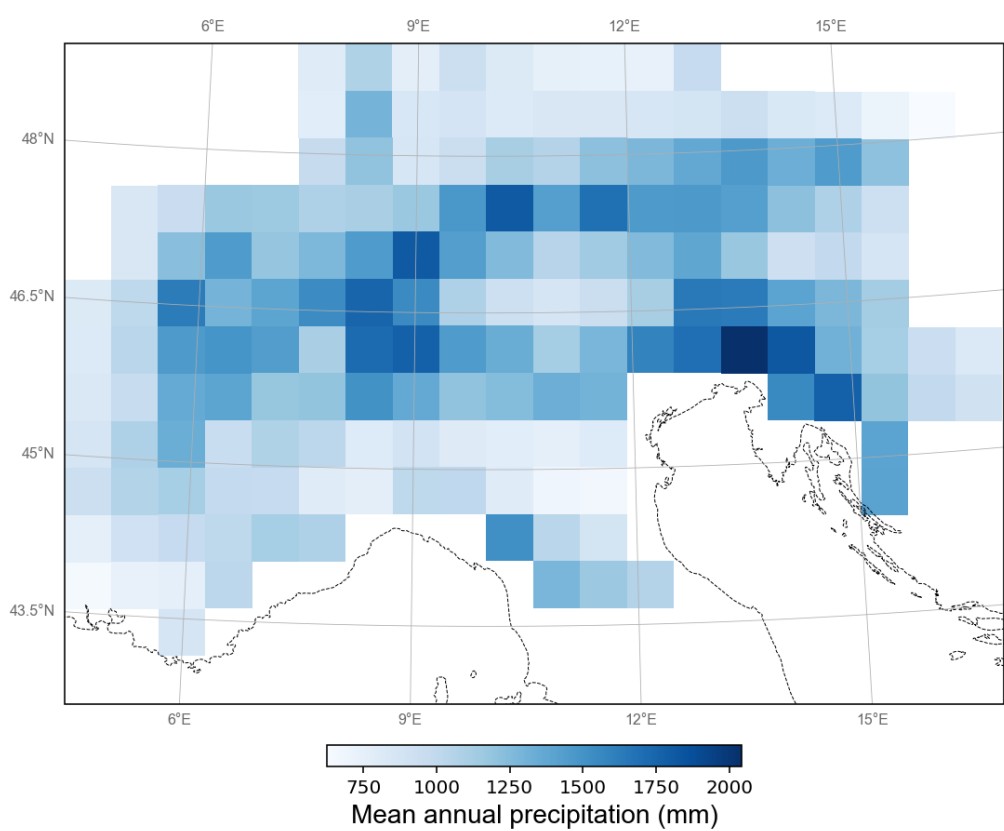

**Figure A1.** Average annual precipitation of the coarse resolution precipitation fields, obtained by conservatively interpolating the native APGD to a 50km Cartesian grid.





| Parameter | Unit | Role | Parametrisation P1 | Parametrisation P2 |
|---|---|---|---|---|
| $\beta$ | dimensionless | Link between TEI and downscaled precipitation | 16 | 32 |
| $\gamma$ | dimensionless | Coefficient associated with the TEI drying effect correction | $5e^{-10}$ | $5e^{-9}$ |
| $dw\_search$ | kilometers | Windward searching distance used to compute TEI | 60 | 60 |
| $dw\_search_{drying\_effect}$ | kilometers | Windward searching distance used to compute the TEI drying effect correction | 60 | 60 |
| $nbr\_wdir$ | dimensionless | Number of prevailing wind directions | 30 | 30 |
| $max\_P\_increase\_factor$ | dimensionless | Maximum value of $P_{HR}/P_{CR}$ | 100 | 4 |

**Table A1.** Set of parameters of the model. The P1 parametrisation was used to generate results presented in section 4.2, while P2 was applied to obtain downscaled data shown on Fig.9.







| | | |
|---|---|---|
| 1 : Feldkirch | 7 : Milan (Lombardia) | a : Plain |
| 2 : Grenoble | 8 : Pizzo di Coca summit | b : Windward exposure |
| 3 : Lago Maggiore | 9 : Sion | c : Summit |
| 4 : Lausanne | 10 : Turin (Piemonte) | d : Isolated valley |
| 5 : Lucerne | 11 : Triglav summit | e : Leeward exposure |
| 6 : Merano (Sud-Tyrol) | | |

**Figure A2.** Digital Elevation Model covering the European Alps at a 5km resolution and used as input for several results presented in section 4.2.



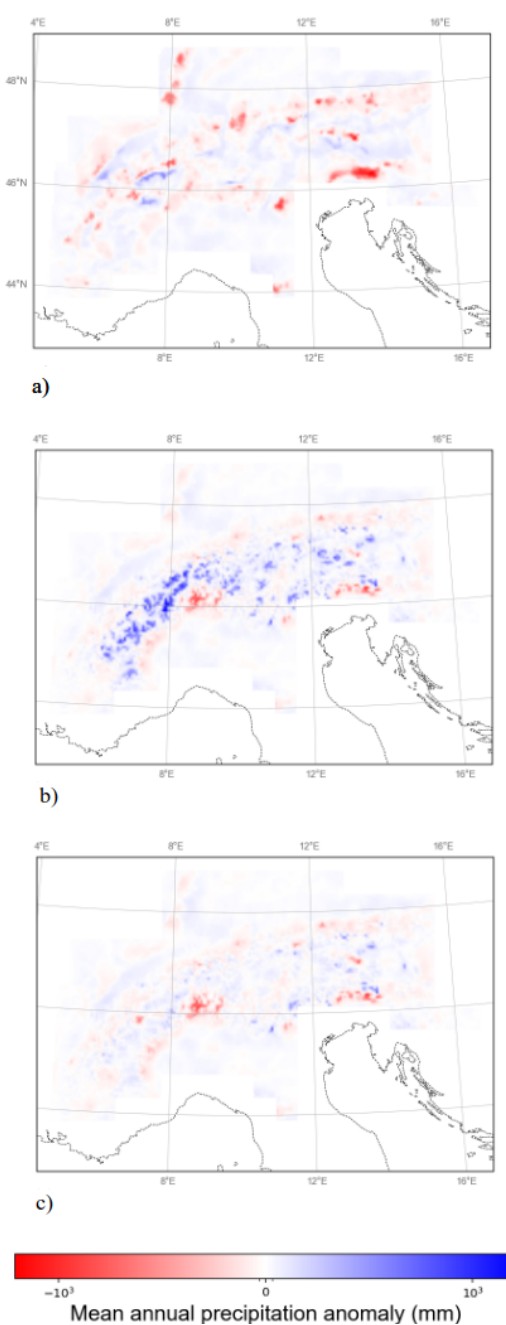

**Figure A3.** Mean annual precipitation anomaly between the native APGD and a) the $Ref_{interpolated}$ dataset b) the $DS$ dataset using GeoDS with the drying effect deactivated, c) the $DS$ dataset using GeoDS with the drying effect activated, calculated over the period 1971-2019 and plotted using a logarithmic scale of colours. Red (respectively blue) areas correspond to underestimated (resp. overestimated) predicted precipitation compared to the target. The $DS$ datasets were generated using the P1 parametrisation (Appendix A Table A1).





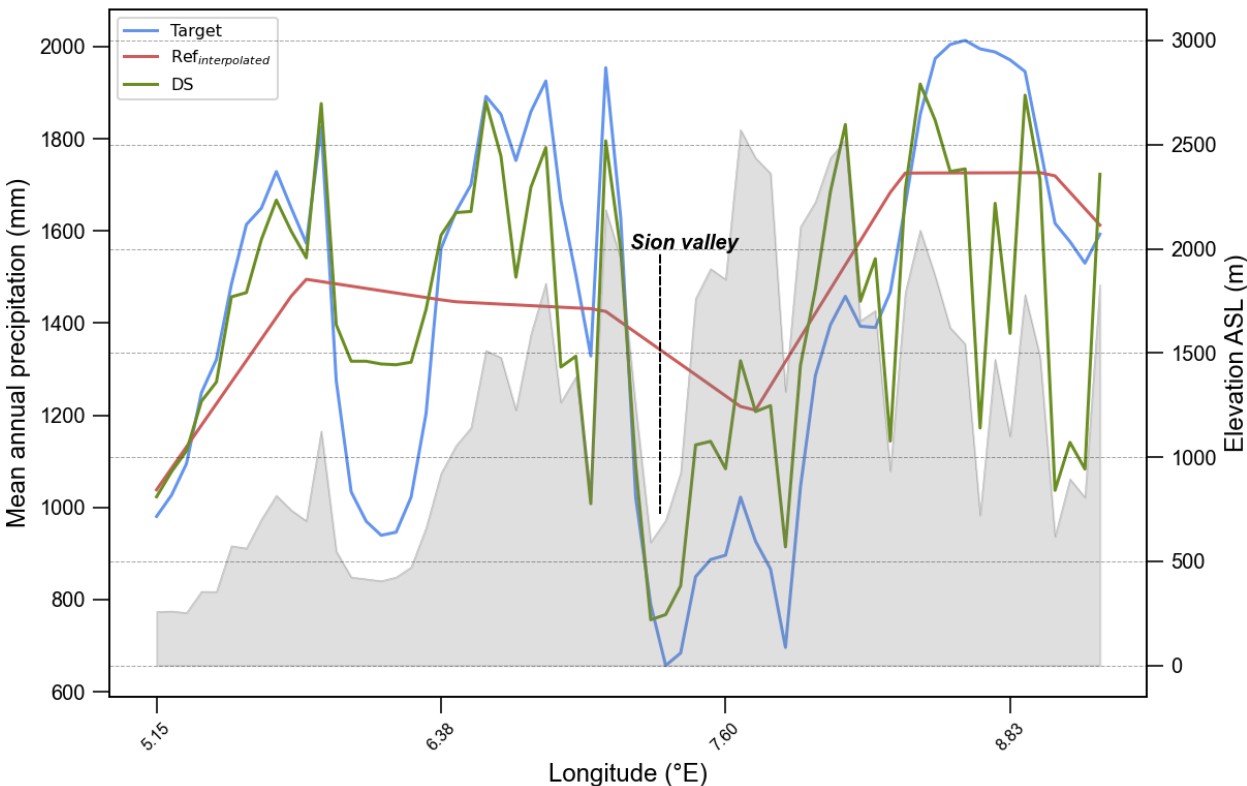

**Figure A4.** Mean annual precipitation (colored lines) and topographic elevation (filled grey curve) along a longitudinal section across the Alps (B segment on Figure 4.c)



**Figure A5.** Mean target and predicted precipitation interannual variability calculated over the period 1971-2019 on a global scale and for particular topographical contexts



*Author contributions.* JBB AQ DR and DP designed the project. JBB developed the model code and performed the simulations. JBB AQ
DR DP and PVA analysed the results. JBB prepared the manuscript with contributions from all co-authors.

*Competing interests.* The authors declare that they have no conflict of interest.

*Acknowledgements.* We thank the Federal Office of Meteorology and Climatology MeteoSwiss for providing the Alpine Precipitation Gridded Dataset used in this study. Our research was conducted at the Laboratoire des Sciences du Climat et de l'Environnement, LSCE/IPSL, CEA-CNRS-UVSQ, Université Paris-Saclay, Gif-sur-Yvette, France.





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
