# Peer review of "GeoDS (v.1.0) : a simple Geographical DownScaling model for long-term precipitation data over complex terrains"

_EGUsphere, 2025_

## Referee Comment (RC1)

**Review of 'GeoDS (v.1.0): a simple Geographical DownScaling model for long-term precipitation data over complex terrains'**

**Link to manuscript.**

In general, I found the paper difficult to read and digest because of the way it's written. It does provide a lot of technical details, but I get lost in them. Making use of unfamiliar acronyms and abbreviations doesn't help. It's funny, as the mathematical notation spells out their representation in the form of subscripts. Nevertheless, after having read the paper, I'm struggling to remember what it was all about and what were the important points. The paper also contains the odd graphics without reference nor caption. Perhaps it's because of the way the paper was structured, but I also think the paper is not ready for submission because it still bears the character of being in the draft stage. I got half way through, and thought it would be a struggle to get through the rest. I suggest a major revision and then that the authors make sure that it's really ready for the next revision.

**Abstract**

The abstract makes a number of undocumented statements and resembles more an introduction than an abstract. It has to be rewritten to provide an overview on what was done and what was achieved, and statements about the science should be moved to the introduction and should come with proper references.

The phrase "order of tens of kilometers" - the models have a minimum skillful scale (see e.g. DOI:10.2151/jmsj.2015-042) of around 8 grid-boxes, and the global climate models typically have a spatial resolution of ~100 km, so tens of kilometres is not right.

"For climate applications, it is notoriously difficult to generate high spatial resolution data over long timescales (typically millennial)" - not really with empirical-statistical downscaling calibrated on station data, for instance represented through principal component analysis (PCA) components, and then gridded over a limited region. But it's too computationally costly with dynamical downscaling. See e.g. DOI:10.5194/hess-29-45-2025.

The statement "but they often exhibit important limitations when applied over long periods of time" is strange and warrants at least a reference. But I don't see the limitations that the authors do, because empirical-statistical downscaling requires little computational resources and can be applied to millennia-long simulations.

The real test is to assess the skill in terms of predicting changes over time - spatial variations are not so hard and often encoded in the local in-situ measurements.

**Introduction**

The sentence "..., limitations regarding their spatial and temporal cover-age often lead to combining them with data derived from models" doesn't make sense.

The numerical models have a minimum skillful scale associated with their design which is larger than the model resolution, see e.g. doi:10.2151/jmsj.2015-042. The introduction could give a fuller account on downscaling and explain what downscaling is, as Al-based methods have lots to learn from empirical-statistical downscaling (ESD).

The sentence "Such limited-area models are supposedly able to spatially and temporally refine the global circulation signal by physically accounting for sub-grid processes and effects" may be a bit misleading - it's often desired that the regional climate models (RCMs) reproduce the global circulation signal provided by the global climate model (GCM), and are therefore often subject to spectral nudging to ensure physical consistency between the two. The RCMs provide an improved representation of the topography and do the calculations on a finer grid.

All models rely on the stationary assumption, e.g. the parameterisation schemes which upscale unresolved (e.g. cloud microphysics) to a greater volume. Hence the statement "SDS relies on a strong hypothesis of stationarity" gives a misrepresentation that this only is a caveat for ESD, which is far from the truth. The question of stationarity can be deal with by downscaling the parameters describing the shape of statistical distributions rather than day-by-day outcome and it is possible to use kriging with elevation as a covariate to get results for a whole area in addition to the sites where measurements have been made (e.g. DOI:10.5194/hess-29-45-2025).

The rain gauge data from different locations embed information about the effect of geographical factors such as slope and elevation as well as large-scale ambient conditions or teleconnection. It is therefore also possible to downscale a group of rain gauge records and subsequently use kriging to model the effect on topography, rather than trying to use topography-based models to try to predict the rainfall at different locations. It's also useful to keep in mind that statistical properties are more systematically dependent on geographical factors and vary more predictable in space than more random daily or monthly outcomes.

The use of many different variables at different levels from a climate model may provide a tight fit (over-fit?), but also places very strict demands that the model used for projection/prediction reproduces the internal structure between these so that the downscaling makes sense. The internal structure may change as a consequence of climate change/variability, and it's not obvious which variable or level should carry most weight if they diverge. This is the reason why 'downscaling weather' (predicting the state for each time step) may not be such a good idea for climate studies (but good for forecasting) and 'downscaling climate' (i.e. statistical parameters) based on univariate predictors may be a more robust method.

**Methods**

A focus is on orographic precipitation, but it's important to regard precipitation as a byproduct of a range of different meteorological phenomena: cyclones, cut-off lows, fronts, convection, atmospheric rivers as well as orographic forcing (some of them may overlap or be related). There is both warm and cold initiation of precipitation (ice crystals, cloud drops, snow and rain) depending on the situation. The description of meteorological situations should make use of references.

L159: typo "]- $\pi$ ,  $\pi$ ]". Equation 1 is difficult to understand - what is h() and how does w factor in? Equation 2 makes me wonder if it has been Al-generated with such elaborate naming? Figure 2: show the prevailing wind as well as arrows?

**Precipitation downscaling**

One question is whether the GCMs reproduce the prevailing wind with realism. The GCMs' ability to reproduce the winds needs to be evaluated. The limitation of this type of downscaling is that it only reproduces orographic type precipitation.

Fig. 6 shows a surprisingly good match between downscaled and observed monthly precipitation. If the downscaled results only captures orographic precipitation, that is even more surprising, as I'd expect rain from thunder storms over the Alps from time to time.

I stopped here because I felt that the paper was not ready for submission.

---

## Author Comment (AC1)

**Authors' response to the comments of the first reviewer**

**1/ General response :**

Although we generally agree with the scientific comments of the reviewer, we believe that most of them are not applicable to this specific study. The value of our method needs to be apprehended in the light of what long-term simulations may involve in terms of climate changes compared to the present/recent past. Paleoclimates cover a wide range of Earth system configurations, with characteristics often radically different from what we know today : altered geography and topography due to plate tectonics, modified orbital parameters leading to changes in seasonal cycles, major differences in CO2 levels, etc. Although techniques like Empirical Statistical Downscaling (ESD) are valuable downscaling tools, they are highly constrained by the observational dataset used for their calibration. The empirical relationships between large scale predictors and local predictands, typically built using recent past observations, are not guaranteed to remain valid through time, especially under very different climate conditions. The limits to such approaches can be easily illustrated when downscaling General Circulation Model (GCM) simulations during the Last Glacial Maximum for instance (~21 ky. ago), as it is a common case study in paleoclimatology. Among other characteristics, this period is associated with a global mean temperature 4.5 +/- 0.9 degrees lower than today (Annan et al., 2022, doi.org/10.5194/cp-18-1883-2022), an altered oceanic circulation (Lynch-Stieglitz et al., 2007, DOI: 10.1126/science.1137127), a mean sea level about 120m lower than today (Peltier and Fairbanks, 2006, doi.org10.1016/j.quascirev.2006.04.010) and the presence of massive ice sheets (Laurentide, Fenoscandian) covering a large part of North America and Western Eurasia. At this time, a large ice field was covering the European Alps (doi.org/10.1038/s41467-025-56168-3) and the dominant circulation patterns were likely different from today, dominated by southwesterly moisture advection (Becker et al. 2016, doi:10.5194/gh-71-173-2016). In such contexts, where global and local climate as well as topography differ from today's, there is a need for robust and cost-effective downscaling approaches, since relying on empirical relationships derived from present observations may lead to erroneous results, while using dynamical methods remains computationally expensive.

The method proposed in this paper is not about simulating local climate conditions in a meteorological way (downscaling weather is not the purpose of the study and cannot be dealt with in paleoclimate contexts), but instead to capture efficiently dominant regional patterns using topography-based information. The reason the method focuses on orography to downscale climate is that it is a major driver of local variability in mountainous areas, while being a relatively accessible information for various paleoenvironments through terrain data. We agree with the reviewer to say that physics resolving models such as regional climate models (RCM) also contain parametrisations calibrated upon recent past observational datasets. As such, the stationarity assumption also affects physically-based approaches. This aspects is also why we have decided to represent only one key process related to precipitation, i.e. the impact of orography. In any climatic context (paleo or long-term future) the relationships on which we based our approach are expected to be preserved (namely, windward increase and leeward decrease of precipitation due to climate/orography interactions).

For these reasons, while most remarks in the review are true we believe that they are out of the scope of the paper. For clarity concerns, the revised version of the manuscript will insist further on the motivations and the purpose of the method since we are aware that our approach differs from the techniques most commonly used in the literature.

Please find below the detailed responses of the authors to the reviewer's comments (indicated in italic). Corrections added to the revised manuscript are indicated in bold blue.

**2/ Detailed response :**

General comments :

1) « *In general, I found the paper difficult to read and digest because of the way it's written. It does provide a lot of technical details, but I get lost in them* »

**[Response]** We would appreciate specific examples as we would gladly improve our manuscrit. The english vocabulary as well as the structure of the sentences respect academic research standards. The reviewer is welcome to provide an exhaustive list of langage issues that could justify this comment. As such, our manuscript conforms with GMD standards which explictly state : « The main paper should describe both the underlying scientific basis and purpose of the model and overview the numerical solutions employed. The scientific goal is reproducibility: ideally, the description should be sufficiently detailed to in principle allow for the re-implementation of the model by others, so all technical details which could substantially affect the numerical output should be described. Any non-peer-reviewed literature on which the publication rests should be either made available on a persistent public archive, with a unique identifier, or uploaded as supplementary information. »

2) « *Making use of unfamiliar acronyms and abbreviations doesn't help.* »

**[Response]** Using acronyms is allowed, as long as they are explained in the text. Several of them are of classic use within the climate scientific community. This is the case for GCM (Global Climate Model), RCM (Regional Climate Model), or DEM (Digital Elevation Model). DDS (for dynamical downscaling) is used for instance in one of the articles the reviewer refers to (DOI:10.2151/jmsj.2015-042), and SDS, for statistical downscaling, follows a similar logic. GDS (for Geographical Downscaling) is indeed less common, but once again, it follows the same nomenclature. GeoDS is the name we gave to our model and remains quite simple, as it stands for « Geographical DownScaling » (L120). APGD is the name given by Isotta et al. to the climatology they built over the Alps (see section 2.1.1). The index m (e.g. L117) stands for monthly, as explained in the paper.

The model has several parameters, for which a summary table is available in the Appendix section (table A1). As for the rest of the variables, they are explained every time they are used for the first time.

*3) « Nevertheless, after having read the paper, I'm struggling to remember what it was all about and what were the important points »*

**[Response]** The objectives of the paper are clearly stated where they can be expected : in the abstract (L6-7) ; at the end of the introduction (89-90-91) ; at the end of the conclusion (L508-513). That being said, the introduction was revised to further highlight the difficulty inherent to multi-millenial, palaeoclimate downscaling.

**[Revised version]** : « **Additionally, SDS relies on a strong hypothesis of stationarity, supposing that the statistical relationship built on the observation period (generally 50 to 150 years) remains valid through time (Shoof, 2013 ; Ludwig et al. 2019). Although this assumption is reasonable for recent past and present, it may not stay valid in paleoclimate contexts which often involve major climate changes compared to the calibration period (Karger et al. 2023 ; Arthur et al. 2023). For instance, the Last Glacial Maximum (a classic case study in paleoclimatology) was associated, among other characteristics, with a global mean temperature 4.5 +/- 0.9 degrees lower than today (Annan et al. 2022), an altered oceanic circulation (Lynch-Stieglitz et al. 2007), a mean sea level about 120m lower than today (Peltier and Fairbanks, 2006) and the presence of massive ice sheets (Laurentide, Fenoscandian) covering a large part of North America and Western Eurasia. In such contexts, where both global and local climate characteristics, as well as surface features (e.g. topography, land cover) differ from today's, the statistical relationships between large scale predictors and local predictands are not guaranteed to be preserved. While stationarity issues are not restricted to SDS approaches, these are assumed to be less robust over time than physically-based relationships.** »

*4) « The paper also contains the odd graphics without reference nor caption »*

**[Response]** Several captions were indeed cut off during formatting. We apologise for this and will correct it for the next revision.

*5) « Perhaps it's because of the way the paper was structured, but I also think the paper is not ready for submission because it still bears the character of being in the draft stage. »*

**[Response]** The structure of the paper (Introduction and presentation of the general context, listing of the datasets used in the study, presentation of the method, results and discussion) is quite common and respects GMD standards. We consider the comment comparing the paper to a draft as an unjustified and questionable criticism. The article went through an access review performed by the handling editor and was accepted for online discussion, which would not have been the case for a draft. Except minor details which will naturally be corrected (e.g. captions being cut off), the paper respects writing, structure and documentation standards expected from a scientific article. The reviewer is welcome to justify this remark with proper examples.

Abstract :

*6) « The abstract makes a number of undocumented statements and resembles more an introduction than an abstract. It has to be rewritten to provide an overview on what was done and what was achieved, and statements about the science should be moved to the introduction and should come with proper references. »*

**[Response]** It is common for an abstract to give a quick overview of the general context of the paper (in this case : 6 lines out of 20). References are not mandatory, and statements appearing in the abstract are selected from the introduction, where they are carefully documented with proper references. The following article is one of GMD's highligh papers and follows the same logic (https://doi.org/10.5194/gmd-18-7035-2025). Contrary to what is suggested by the reviewer, the overview of what was done and what was achieved in this study is available in the abstract (L6-20).

*7) « The phrase "order of tens of kilometers" - the models have a minimum skillful scale (see e.g. DOI:10.2151/jmsj.2015-042) of around 8 grid-boxes, and the global climate models typically have a spatial resolution of ~100 km, so tens of kilometres is not right. »*

**[Response]** In the abstract, and in the introduction L30, we refer to the spatial resolution of the GCM, not their skillful scale. It is today possible to run GCM over several decades at a spatial resolution of the order of tens of kilometers (e.g. doi.org/10.1029/2020MS002298 or refer to the High-Resolution Model Intercomparison Project). We yet acknowledge that for long-term simulations, models indeed exhibit coarser resolutions. This point was clarified in the revised version.

**[Revised version]** : « Global climate models offer the most comprehensive description of the climate system and its internal processes to date but current computational capabilities typically restrict their spatial resolution to the order of hundreds of kilometers when long-term (millennial or longer) simulations are needed. »

*8) « "For climate applications, it is notoriously difficult to generate high spatial resolution data over long timescales (typically millennial)" - not really with empirical-statistical downscaling calibrated on station data, for instance represented through principal component analysis (PCA) components, and then gridded over a limited region. »*

**[Response]** Climate applications can indeed be apprehended in various manners. This is the reason why we explain in the introduction that this work was motivated by the need for a robust and low resources demanding method adapted to long-term climate studies, likely to involve major climate changes. The pertinence of an ESD calibrated on station data (recent climate conditions) and applied e.g. over the Alps during the LGM, with the presence of an ice-sheet and an entirely different climate system, is debatable. Using for instance the PCA would rely on the strong hypothesis that the prevailing modes of climate variability are stationary between the two periods. On the other hand, relying on the description of simple physical processes (orographic uplift, leeward rain-shadow effect) to spatially distribute precipitation is expected to remain valid through time, as long as large-scale inputs and fine scale-topography are correct (strong working hypothesis discussed below).

*9) « But it's too computationally costly with dynamical downscaling. »*

   **[Response]** We agree with this statement (see L41-42).

*10) « The statement "but they often exhibit important limitations when applied over long periods of time" is strange and warrants at least a reference. But I don't see the limitations that the authors do, because empirical-statistical downscaling requires little computational resources and can be applied to millennia-long simulations. »*

   **[Response]** Downscaling long-term climate simulations often involves dealing with major climate changes while being limited by computational resources. Consequently, specific issues arise when using classic downscaling methods (SDS and DDS). Limitations of ESD methods in relation to the objective of the study are developped in the general response hereinabove.

*11) « The real test is to assess the skill in terms of predicting changes over time - spatial variations are not so hard and often encoded in the local in-situ measurements. »*

   **[Response]** Spatial variations under entirely different climate conditions (refer to the general response above) are not as simple to predict as suggested. As for changes over time, it is not meant in a meteorological way (i.e. Rreconstructing past weather). The purpose of the method is to deal with temporal changes from a climatological perspective, i.e. how spatial patterns evolve with changing topography and utterly different climate forcings for instance.

Introduction :

*12) « The sentence "..., limitations regarding their spatial and temporal coverage often lead to combining them with data derived from models" doesn't make sense »*

   **[Response]** We ask the reviewer to provide specific examples of their comment. Since obviously observational proxy are spatially limited and only cover certain periods, data from models are needed to conduct various applications.

*13) « The numerical models have a minimum skillful scale associated with their design which is larger than the model resolution, see e.g. doi:10.2151/jmsj.2015-042. The introduction could give a fuller account on downscaling and explain what downscaling is, as AI-based methods have lots to learn from empirical-statistical downscaling (ESD) »*

   **[Response]** The description of the downscaling covers only aspects relevant to the purpose of the study. The topic of the paper is not about AI-based methods.

*14) « The sentence "Such limited-area models are supposedly able to spatially and temporally refine the global circulation signal by physically accounting for sub-grid processes and effects"*

*may be a bit misleading - it's often desired that the regional climate models (RCMs) reproduce the global circulation signal provided by the global climate model (GCM), and are therefore often subject to spectral nudging to ensure physical consistency between the two. The RCMs provide an improved representation of the topography and do the calculations on a finer grid. »*

**[Response]** We agree with the reviewer that an increased resolution does not necessarily involve explicitly accounting for additional processes. A RCM can refine the forcing GCM simply because of a more detailed orography, allowing to better represent atmospheric dynamics. This was misleading and was rephrased in the revised version.

**[Revised version]** : « **Such limited-area models are supposedly able to refine the global circulation signal by solving fluid dynamics on a finer spatial grid (e.g. with a more detailled topography) and by potentially accounting for additional processes.** »

*15) « All models rely on the stationary assumption, e.g. the parameterisation schemes which upscale unresolved (e.g. cloud microphysics) to a greater volume. Hence the statement "SDS relies on a strong hypothesis of stationarity" gives a misrepresentation that this only is a caveat for ESD, which is far from the truth. The question of stationarity can be deal with by downscaling the parameters describing the shape of statistical distributions rather than day-by-day outcome and it is possible to use kriging with elevation as a covariate to get results for a whole area in addition to the sites where measurements have been made (e.g. DOI:10.5194/hess-29-45-2025). »*

**[Response]** As indicated in the general response above, we agree that the hypothesis of stationnarity is not restricted to ESD, since even RCM rely on parametrisations typically calibrated on recent past observations. However, physically-based relationships are assumed to be more robust in time.

An interpolation technique like kriging would require a constant topography through time. This is usually not the case for paleoclimate studies (e.g. Presence of the alpine ice sheet during the LGM).

*16) « The rain gauge data from different locations embed information about the effect of geographical factors such as slope and elevation as well as large-scale ambient conditions or teleconnection. It is therefore also possible to downscale a group of rain gauge records and subsequently use kriging to model the effect on topography, rather than trying to use topography-based models to try to predict the rainfall at different locations. It's also useful to keep in mind that statistical properties are more systematically dependent on geographical factors and vary more predictable in space than more random daily or monthly outcomes. »*

**[Response]** Techniques applicable to the present with (often) well distributed observation data are not necessarily transferable to other contexts. Paleoclimate applications need to deal with limited terrain data.

*17) « The use of many different variables at different levels from a climate model may provide a tight fit (over-fit?), but also places very strict demands that the model used for projection/prediction reproduces the internal structure between these so that the downscaling makes sense. The internal*

*structure may change as a consequence of climate change/variability, and it's not obvious which variable or level should carry most weight if they diverge. This is the reason why 'downscaling weather' (predicting the state for each time step) may not be such a good idea for climate studies (but good for forecasting) and 'downscaling climate' (i.e. statistical parameters) based on univariate predictors may be a more robust method »*

**[Response]** We generally agree with this comment. However, we are not trying to predict weather, but to capture major climatological patterns (monthly timestep) from topographical information and limited GCM outputs assuming that the physically-based description of the interactions between large-scale circulation and orography remains valid through time.

Method :

*18) « A focus is on orographic precipitation, but it's important to regard precipitation as a byproduct of a range of different meteorological phenomena: cyclones, cut-off lows, fronts, convection, atmospheric rivers as well as orographic forcing (some of them may overlap or be related). There is both warm and cold initiation of precipitation (ice crystals, cloud drops, snow and rain) depending on the situation. The description of meteorological situations should make use of references. »*

**[Response]** The discretisation of the different meteorological situations occuring in mountainous areas would require a potentially wide range of GCM variables used as co-variables to downscale precipitation. Besides the challenges regarding the physical integration of each co-variables to the model (weight, relationships between them, etc...), many GCM outputs for paleoclimate simulations are available at a too coarse temporal resolution (e.g. monthly) to achieve such distinctions.

*19) « Equation 1 is difficult to understand - what is h() and how does w factor in? »*

**[Response]** Equation 1 is the equation used to compute the topographic exposure index (noted quite logically TEI) of any gridpoint M, and for any prevailing wind direction w. We will not explain every term of the equation here, as they are clearly defined in the paper, right under equation 1 (h(M) among other factors, L180-181). We acknowledge that the index w on the left side of the equation can be misleading, as it does not explicitly appears in the right term of Eq.1. Initially, we made this decision in order to avoid overcomplicating the equation with multiple indexes. We added a w index to the right term of the equation in the revised version for clarity concerns.

*20) « Equation 2 makes me wonder if it has been AI-generated with such elaborate naming? »*

**[Response]** Equation 2 describes how the correction of the TEI for any gridpoint M and used to mimic the drying effect occuring over large mountain ranges is computed. Since this is a methodological paper, it seemed important to be as explicit as possible when naming the different

variables used in the model. As indicated when submitting the paper to GMD, we did not make use of AI to generate the code of the model nor the equations it is based on.

*21) « Figure 2: show the prevailing wind as well as arrows? »*

[Response] Arrows indicating the prevailing winds can indeed improve the clarity of this figure and were added in the revised manuscript (see below).

[Figure]

Precipitation downscaling :

*22) « One question is whether the GCMs reproduce the prevailing wind with realism. The GCMs' ability to reproduce the winds needs to be evaluated »*

[Response] This is a pertinent observation, as the method relies to a large extent on large-scale winds to spatially distribute fine-scale precipitation. A strong working hypothesis of this study is that the source GCM fields to be downscaled reproduces accurately the atmospheric circulation characteristics at its own resolution (see lines 146-147). In other words, GeoDS is meant to be used on debiased models. In our case, to mimic debiased GCM precipitation fields, we upscaled the APGD to a 50km cartesian grid (see section 2.1.2). For the wind, we selected ERA5 u and v fields, for this dataset is commonly used as a climate reference for the recent past. Without observational data to correct ERA5 biases, we relied on the quality of the resulting downscaled precipitation fields (see section 4) to evaluate the quality of the reanalysis wind fields. We also conducted tests at a daily timestep to further investigate winds effects on the model's performances.

It is naturally expected for the method to perform less well with GCM inputs than with ERA5. We initiated a discussion regarding this matter in section 5.2. Since this paper was written to present our method, its advantages and limitations, it seemed pertinent to limit biases from the inputs used, rather than focusing on GCM inherent biases, which would lead to a entirely different study. A

paleoclimate study, presenting downscaled precipitation fields of Paleoclimate Modelling Intercomparison Project (PMIP) models with GeoDS, is currently an other work in progress.

23) « *The limitation of this type of downscaling is that it only reproduces orographic type precipitation.* »

**[Response]** It is indeed an important limitation of the method. However, as shown by the different results presented in this study, windward orographic uplift and condensation of moist air masses, associated with leeward rain-shadow effect control to a large extent spatial patterns of precipitation in complex environments (see references within paper, e.g. Roe 2004 or Smith 1979). We are perfectly aware that our model does not take into account every aspect of precipitation in mountainous areas, as noted in section 6. However, our goal (as indicated multiple times in the article : L89, 139, 509-513) is to propose a simple and fast running procedure to capture the dominant effect of the climate/orography interactions.

24) « *Fig. 6 shows a surprisingly good match between downscaled and observed monthly precipitation. If the downscaled results only captures orographic precipitation, that is even more surprising, as I'd expect rain from thunder storms over the Alps from time to time.* »

**[Response]** These results can be reproduced with the codes and data stored under the indicated Zenodo archive (See section *Code and data availibility*). They illustrate the dominant effect of orographic precipitation at the time scale considered. The remark of the reviewer regarding storms is however very pertinent, as it touches upon a critical aspect of precipitation in the Alps and a limit of our model. By construction, GeoDS focuses on orographic precipitation and does not solve local convection, essentially occuring during summer. Although the downscaling works well annually, decreased performances could be expected during summer. To assess the robustness of the method seasonnally, we analysed the climatologies and quantile-quantile plots of winter and summer precipitation over the whole period. Results are presented hereafter.

[Figure]

*Figure 1* : *Average winter (DJF, left panel) and summer (JJA, right panel) precipitation for the Ref_{interpolated} (a), the GeoDS downscaled (b) and the native high resolution (c) versions of the Alpine Precipitation Grid Dataset over the period 1971-2019 using a 10km input DEM.*

[Figure]

*Figure 2* : *Comparison of winter (DJF) observed precipitation quantiles and a) Ref$_{interpolated}$ b) downscaled with GeoDS precipitation quantiles using a 10km input DEM.*

[Figure]

*Figure 3* : *Comparison of summer (JJA) observed precipitation quantiles and a) Ref$_{interpolated}$ b) downscaled with GeoDS precipitation quantiles using a 10km input DEM.*

The algorithm performs better during summer than winter, both statistically and in term of spatial distribution of precipitation. Besides biases in regions like the Slovenian Alps (discussed in the paper), the model exhibits an important dry bias in the Austrian Alps (north-east edge of the massif) during winter. Although these results may be counterintuitive at first glance, they can be explained by taking the following elements into account. A first explanation lies in how the large-scale precipitation data to downscale were built for this study. As explained in section 2.1.2, we upscaled the native APGD dataset to a regular 50km Cartesian grid to obtain a pseudo GCM output. This allows to feed GeoDS with debiased data and ensures that errors between downscaled precipitation patterns and the observations are primarily caused by the downscaling method, rather than propagating from the large-scale model (naturally, biases are expected to come from ERA5 winds as well. However, we lack an observational dataset to correct them). Since the APGD is derived from a well-distributed rain-gauge stations network, it captures precipitation patterns associated with local convection. Although upscaling the APGD smoothes this summer signal, it still appears in the large-scale data used as GeoDS inputs, and therefore, in the downscaled precipitation fields as well.

A second potential reason for the good model-data agreement during summer is that moisture feeding convective precipitation is primarily advected, rather than locally recycled. Using ERA-40 data, Sodemann and Zubler (2009) estimated the beta ratio (defined as the fraction of precipitation inside a domain that originates from evaporation inside the same domain, Schär et al., 1999) over the Alps to about 15.5 % for summer (below 1% during winter). So, although convection of local evaporation is not negligible, most summer moisture (primarily coming from land evaporation) still needs to be advected through large-scale circulation. Even if the dominant summer driver of condensation may be thermic convection rather than classic orographic uplift, the model is still able to consistently distribute fine-scale precipitation over the domain based on monthly average wind, indicating where moisture primarily comes from.

When analysing the seasonal performances of the model, it should be kept in mind that we set beta and delta values (dimensionless parameters that modulate the effect of topography on fine-scale precipitation) based on 1) the optimisation of several statistical indicators 2) the comparison of precipitation spatial patterns on an annual basis. Investigating a seasonal parametrisation of GeoDS may help improving the model's performances, especially during winter, for which the model exhibits a dry bias.

**[Revised version]** The revised version of the manuscript will include a paragraph regarding the seasonal performances of the model.

25) « *I stopped here because I felt that the paper was not ready for submission.* »

**[Response]** It is confusing to note that the reviewer decides to stop the reading where he/she does, despite results worth discussing. The whole discussion was ignored.

---

## Author Comment (AC2)

**Authors' response to the comments**
**of the second reviewer**

We thank the reviewer for the time and efforts put into reviewing our manuscript and for the valuable remarks made. Please find hereafter the reviewer's comments (in italic), and our responses. Corrections added to the revised manuscript are indicated in bold blue.

1) « *The main idea that local precipitation is mainly driven by the amount of large-scale precipitation and the direction of wind relative to the topography seems to be applicable only for large-scale precipitation, but certainly not so for convective precipitation. The area where the methodology is being tested, Switzerland, experiences both types of precipitation, mainly depending on the season. The analysis of the method's skill is not seasonally stratified, so my concern is whether this skill is primarily derived from winter-time large-scale frontal precipitation. I think a seasonal stratification of the skill needs to be included and possibly discussed if there are seasonal differences. If they are, would the methodology need to include other fields, such as near-surface air temperature or air column stability, to account for convection?* »

**[Response]** Thank you for this comment, as it touches upon a critical aspect of the precipitation regime in the Alps as well as to a limit of our model. We give hereafter the same response we made to one of the first reviewer's comments. By construction, GeoDS focuses on orographic precipitation and does not solve local convection, essentially occuring during summer. Although the downscaling works well annually, decreased performances could be expected during summer. To assess the robustness of the method seasonnally, we analysed the climatologies and quantile-quantile plots of winter and summer precipitation over the whole period. Results are presented hereafter.

[Figure]

*Figure 1* : *Average winter (DJF, left panel) and summer (JJA, right panel) precipitation for the Ref$_{interpolated}$ (a), the GeoDS downscaled (b) and the native high resolution (c) versions of the Alpine Precipitation Grid Dataset over the period 1971-2019 using a 10km input DEM.*

[Figure]

*Figure 2 : Comparison of winter (DJF) observed precipitation quantiles and a) Ref$_{interpolated}$ b) downscaled with GeoDS precipitation quantiles using a 10km input DEM.*

[Figure]

*Figure 3 : Comparison of summer (JJA) observed precipitation quantiles and a) Ref$_{interpolated}$ b) downscaled with GeoDS precipitation quantiles using a 10km input DEM.*

The algorithm performs better during summer than winter, both statistically and in term of spatial distribution of precipitation. Besides biases in regions like the Slovenian Alps (discussed in the paper), the model exhibits an important dry bias in the Austrian Alps (north-east edge of the massif) during winter. Although these results may be counterintuitive at first glance, they can be explained by taking the following elements into account. A first explanation lies in how the large-scale precipitation data to downscale were built for this study. As explained in section 2.1.2, we upscaled the native APGD dataset to a regular 50km Cartesian grid to obtain a pseudo GCM output. This allows to feed GeoDS with debiased data and ensures that errors between downscaled precipitation patterns and the observations are primarily caused by the downscaling method, rather than propagating from the large-scale model (naturally, biases are expected to come from ERA5 winds as well. However, we lack an observational dataset to correct them). Since the APGD is derived from a well-distributed rain-gauge stations network, it captures precipitation patterns associated with local convection. Although upscaling the APGD smoothes this summer signal, it still appears in the large-scale data used as GeoDS inputs, and therefore, in the downscaled precipitation fields as well.

A second potential reason for the good model-data agreement during summer is that moisture feeding convective precipitation is primarily advected, rather than locally recycled. Using ERA-40 data, Sodemann and Zubler (2009) estimated the beta ratio (defined as the fraction of precipitation inside a domain that originates from evaporation inside the same domain, Schär et al., 1999) over the Alps to about 15.5 % for summer (below 1% during winter). So, although convection of local evaporation is not negligible, most summer moisture (primarily coming from land evaporation) still needs to be advected through large-scale circulation. Even if the dominant summer driver of condensation may be thermic convection rather than classic orographic uplift, the model is still able to consistently distribute fine-scale precipitation over the domain based on monthly average wind, indicating where moisture primarily comes from.

When analysing the seasonal performances of the model, it should be kept in mind that we set beta and delta values (dimensionless parameters that modulate the effect of topography on fine-scale precipitation) based on 1) the optimisation of several statistical indicators 2) the comparison of precipitation spatial patterns on an annual basis. Investigating a seasonal parametrisation of GeoDS may help improving the model's performances, especially during winter, for which the model exhibits a dry bias.

2) « *Regarding the manuscript itself, the paragraphs are really long. This could be improved to help the reader discern the chain of thought and to better locate paragraphs in a second reading. For instance, the introduction contains just one long paragraph (!), but this problem is also present in other sections. As a broad rule of thumb, a paragraph should be devoted to developing only one idea/message.* »

**[Response]** Thank you for bringing this up, as it does indeed make the text difficult to read in several places. This aspect was improved in the revised version of the manuscript.

3) « *Regarding the data description, did the data present gaps? Were they somehow filled? The precipitation data were aggregated to monthly sums. If gaps were present, were those months proportionally rescaled?* »

[Response] The high-resolution precipitation data were provided by MeteoSwiss on a daily basis over the domain shown on Figure 9.c (page 23), from January 1, 1971, to December 31, 2019. No gaps were identified in the original dataset (17 897 daily maps covering the same area). The method used by Isotta et al. (2014) to build this continuous dataset combines : 1) the collect of station measurements accross the whole domain, each undergoing a quality control procedure (checks for coding problems and for spatial consistency, identification of suspicious time series) 2) interpolation steps to estimate precipitation in ungauged areas.

As indicated in the dataset documentation (https://surfobs.climate.coperni-cus.eu/documents/ProdDoc_APGD.pdf), the spatial analysis for a day D is achieved in 4 steps :

« *(1) Spatial interpolation of the climatological mean precipitation measurements for the calendar month of D (reference period 1971-1990);*

*(2) Calculation of relative anomalies of station measurements of D with respect to the climatological mean from step 1;*

*(3) Spatial interpolation of relative anomalies;*

*(4) Multiplication of the resulting anomaly field with the climatological mean field.* »

A detailed description of the dataset and the method used to build it is available in Isotta et al. 2014 (DOI : 0.1002/joc.3794). We did not modify the native data before aggregating on a monthly basis.

Please note that spatial gaps appeared at the edges of the area when generating large-scale data. This was caused by the definition of the 50km grid and the conservative interpolation used for the coarsening : when a target grid point overlapped at least one native cell with NaN value, the whole grid point was converted to NaN. This is why the spatial extent of the domain shown in several figures (e.g. Fig4, Fig7) is smaller than area covered by the original APGD. We acknowledge that the upscaling step led to cutting off several areas of interest from the final analysis (like the Apennines, Italy). However, it allowed us to feed GeoDS with debiased inputs and to isolate errors caused by the downscaling step.

[Revised version] : « ... the efforts made to reduce the risk of systematic underestimates at high elevations. **No gaps were identified in the original dataset (17 897 daily maps covering the same area).** Although given on a 5km scale... »

4) « 'using a first order conservative remapping from the Climate Data Operator package'. *The CDO package offers several remapping options. I guess that in this case, the proper way to coarsen the data is to calculate the average of the high-resolution data within the low-resolution cells and not by interpolation. Was the coarsening conducted so?* »

**[Response]** The coarsening was indeed conducted so, using the cdo remapcon command. This ensures that the large-scale precipitation data of any coarse grid point corresponds to the average area-weighted precipitation of overlapped native APGD cells. A clearer description of the upscaling process was added to the revised version of the manuscript.

**[Revised version]** : « ...we degraded the APGDm target dataset to a regular 50 km Cartesian grid using a first order conservative remapping from the Climate Data Operator package (version 2.4.4). **With this method, the large-scale precipitation data of any coarse grid point corresponds to the average area-weighted precipitation of overlapped native APGD cells**. This ensures spatio-temporal consistency between precipitation fields at low and high resolutions... »

5) *« 'On a global scale, the algorithm...' Global scale sounds strange here. The authors probably mean the regional average. »*

**[Response]** The sentence was rephrased in the revised version of the manuscript.

**[Revised version]** : « **At the regional scale (Fig.8.a), the algorithm is slightly less accurate**...»